# The iron-dependent repressor YtgR is a tryptophan-dependent attenuator of the *trpRBA* operon in *Chlamydia trachomatis*

Nick D. Pokorzynski 1,2, Nathan D. Hatch2, Scot P. Ouellette 2 & Rey A. Carabeo 2✉

The *trp* operon of *Chlamydia trachomatis* is organized differently from other model bacteria. It contains *trpR*, an intergenic region (IGR), and the biosynthetic *trpB* and *trpA* open-reading frames. TrpR is a tryptophan-dependent repressor that regulates the major promoter ($P_{trpR}$), while the IGR harbors an alternative promoter ($P_{trpBA}$) and an operator sequence for the iron-dependent repressor YtgR to regulate *trpBA* expression. Here, we report that YtgR repression at $P_{trpBA}$ is also dependent on tryptophan by regulating YtgR levels through a rare triple-tryptophan motif (WWW) in the YtgCR precursor. Inhibiting translation during tryptophan limitation at the WWW motif subsequently promotes Rho-independent transcription termination of *ytgR*, thereby de-repressing $P_{trpBA}$. Thus, YtgR represents an alternative strategy to attenuate *trpBA* expression, expanding the repertoire for *trp* operon attenuation beyond TrpL- and TRAP-mediated mechanisms described in other bacteria. Furthermore, repurposing the iron-dependent repressor YtgR underscores the fundamental importance of maintaining tryptophan-dependent attenuation of the *trpRBA* operon.

[1] School of Molecular Biosciences, College of Veterinary Medicine, Washington State University, Pullman, WA, USA. [2] Department of Pathology and Microbiology, University of Nebraska Medical Center, Omaha, NE, USA. ✉email: rey.carabeo@unmc.edu

Access to the amino acid tryptophan is an important determinant in the outcome of host-pathogen interactions. When infected cells are stimulated with the proinflammatory cytokine interferon-gamma (IFNγ), the expression of indoleamine-2,3-dioxygenase (IDO1) is induced[1]. IDO1 catabolizes tryptophan to N-formylkynurenine[2], thereby depriving pathogens of host-derived tryptophan. Intracellular bacteria such as *Mycobacterium tuberculosis* and *Francisella tularensis* are insensitive to immunological tryptophan depletion which is dependent on their ability to biosynthesize tryptophan[3,4]. Surprisingly, several obligate intracellular human parasites are natural tryptophan auxotrophs, including the Gram-negative bacterium *Chlamydia trachomatis* (*Ctr*), rendering these pathogens susceptible to IFNγ-mediated tryptophan depletion[5–8]. This raises the intriguing question of how such pathogens respond to tryptophan deprivation to survive.

*Chlamydia trachomatis* is the leading cause of bacterial sexually transmitted infections and preventable blindness worldwide[9,10]. Chlamydiae are distinguished by a unique biphasic developmental cycle that interconverts an infectious elementary body (EB) with a vegetative and non-infectious reticulate body (RB)[11]. *Ctr* primarily invades the mucosal epithelia, with tropism for the anogenital tract (serovars D-K and L1-3) or the ocular conjunctiva (serovars A-C). Untreated genital tract infections can result in significant morbidity in the reproductive tract, particularly in women, where chronic infections can cause pelvic inflammatory disease and ectopic pregnancy[12,13]. Such chronic infections are often initially asymptomatic and therefore remain undiagnosed until sequelae develop[14]. Thus, a more complete understanding of the interface between host and pathogen during chlamydial infections stands to aid the development of better diagnostic and therapeutic tools. Given that the host interrupts nutrient trafficking to *Ctr* as a means of controlling the infection, investigating how the pathogen responds to nutritional stress may provide important clues for addressing these outstanding problems.

Co-evolution with their respective mammalian hosts has resulted in significant genome reduction among the *Chlamydiaceae*[15,16], requiring the bacteria to primarily rely on the host for access to essential nutrients such as lipids, nucleotides, biometals, and amino acids[17]. Consequently, prolonged disruption of nutritionally relevant sub-cellular trafficking pathways of the host results in physiological defects to *Chlamydia*, most notable being the development of an aberrant, "persistent" growth state, in which Chlamydiae are viable but developmentally arrested[18]. Deprivation of several nutrients, including tryptophan[8,19] and iron[20–22], has been shown to induce persistence in *Ctr*, and transient stress is sufficient to induce transcriptional changes[23,24]. Despite residing in an intracellular environment that could buffer *Ctr* from extracellular stress, they appear to maintain the ability to adapt. The robustness of this adaptation and whether *Ctr* can tailor its response to specific stresses is not clear. The paucity of transcription factors identified in *Ctr* so far suggests a limited capacity to respond to stress. It has been assumed that among the few transcription factors retained by *Ctr*, many have limited regulons defined by particular co-regulator specificities such as nutrient dependence[25]. However, this assumption does not account for the possibility of sub-functionalization, which may expand the regulatory activity of transcription factors with ostensibly limited roles in chlamydial development.

Studying the regulation of tryptophan biosynthetic operons has provided insight into fundamental aspects of gene regulation in bacteria. In the model bacterium *Escherichia coli*, the *trpEDCBA* operon is regulated by two complementary mechanisms: TrpR-mediated tryptophan-dependent transcriptional repression[26] and TrpL-mediated tryptophan-dependent *cis*-attenuation[27]. TrpR repression is acutely sensitive to reduced levels of tryptophan, derepressing the operon to induce tryptophan biosynthesis[28]. Attenuation, on the other hand, is relieved after severe tryptophan starvation by ribosomal stalling at a tandem tryptophan codon motif (WW) in the TrpL coding sequence[29]. Ribosome stalling at the WW motif promotes the formation of a transcription anti-terminator in the nascent RNA, permitting transcription of the tryptophan biosynthetic genes. Under tryptophan-replete conditions, ribosomes read through the WW motif and formation of an RNA terminator structure terminates transcription of the operon. This dual regulation greatly increases the dynamic range of *trp* operon expression in *E. coli*, fine-tuning their response to varying concentrations of tryptophan.

*C. trachomatis*, which cannot synthesize tryptophan de novo, possesses a tryptophan "salvage pathway," encoded by the *trpRBA* operon[30]. TrpBA is a functional tryptophan synthase that catalyzes the final step of tryptophan biosynthesis by condensing indole with serine to generate tryptophan[31]. The tryptophan salvage pathway is crucial in replenishing the tryptophan pool when *Ctr* has access to indole[32,33]. The organization of the *trpRBA* operon distinguishes *Ctr* from many other bacteria, with *trpR* encoded in the same operon as *trpBA*, separated only by a 348-base pair (bp) intergenic region (IGR). The chlamydial TrpR orthologue mediates tryptophan-dependent transcriptional repression from the major promoter upstream of *trpR* ($P_{trpR}$)[34,35]. However, TrpR cannot regulate transcription from the IGR[35], despite the observation of a tryptophan-dependent transcriptional start site (TSS) for *trpBA* in this region ($P_{trpBA}$)[34], implying a more complex regulation of the operon.

Unexpectedly, the tryptophan salvage pathway is also iron-regulated in *Ctr*. Following brief iron limitation, *trpBA* is upregulated independently of altered *trpR* expression[23,24]. This regulation is mediated by the iron-dependent repressor YtgR, which binds to an operator sequence within the *trpRBA* IGR to repress transcription initiation from the alternative promoter for *trpBA*[24]. Simultaneously, YtgR bound to the IGR inhibits transcription readthrough from $P_{trpR}$. This likely benefits *Ctr* in the lower female genital tract (LGT), an environment that is both indole-rich and iron-poor. *Ctr* acquires iron by intercepting the vesicular recycling of the holotransferrin-transferrin receptor (TfR) complex[36–38]. However, TfR expression is restricted to the undifferentiated basal layers of the stratified squamous epithelium in the LGT[39]. Thus, *Ctr* may be deprived of iron during infection of the fully differentiated upper layers. Additionally, high levels of the antimicrobial iron-sequestering protein lactoferrin are associated with *Ctr* infections[40–42]. Concurrently in the LGT, colonization by various indole-producing obligate and facultative anaerobes can provide *Ctr* access to the substrate for tryptophan salvage[43]. Regulation of $P_{trpBA}$ by YtgR may therefore perform an important function by linking the availability of iron to the anatomical site where indole is readily available to *Ctr* for tryptophan salvage.

With YtgR binding in the IGR and simultaneously interfering with transcription elongation from $P_{trpR}$ and initiation from $P_{trpBA}$, maximal expression of the *trpRBA* operon during tryptophan starvation would require measures to counteract iron-dependent regulation. We therefore hypothesized that *Ctr* must possess a tryptophan-dependent mechanism for overcoming iron-dependent YtgR repression of *trpBA*. YtgR is encoded as the C-terminal domain of a fusion gene, YtgCR, where the N-terminus encodes an ABC transporter permease, YtgC, from which YtgR is cleaved during infection[44,45]. Expression of YtgR therefore requires complete expression of the upstream YtgC domain. Here, we report that YtgR levels are tryptophan-regulated during translation through a rare motif of three sequential tryptophan codons (WWW) in the YtgC domain.

Tryptophan deprivation reduces the expression of YtgR in a WWW motif-dependent manner in two interrelated ways: first, ribosomes fail to read through the WWW motif and inhibit translation, and second, transcription of the YtgR-encoding sequence downstream of the WWW motif is terminated in a Rho-independent manner following translation inhibition. The reduction in YtgR levels ultimately activates *trpBA* expression from P$_{trpBA}$. This mechanism of regulation is remarkably similar to a mechanism of attenuation, where inhibition of translation at sequential tryptophan codons during periods of tryptophan starvation enables the continued transcription of downstream biosynthetic genes. We propose that YtgR regulation of *trpBA* exemplifies an alternative attenuation strategy in prokaryotes, where a *trans*-acting transcriptional regulator is subjected to regulation by tryptophan availability. This allows YtgR to customize its regulon in response to iron and/or tryptophan starvation. With YtgR functioning as a dual biosensor of iron and tryptophan, it is positioned to integrate the chlamydial stress response to two nutrients central to host nutritional immunity.

## Results

### Tryptophan limitation disables iron-dependent repression of *trpBA* expression

If tryptophan limitation inactivated both TrpR and YtgR repression of *trpBA*, we expect maximal expression of *trpBA* during tryptophan starvation. A corollary is that the effect of iron limitation, which only inactivates YtgR repression, would result in elevated but not maximal transcription of *trpBA*. We tested these hypotheses by analyzing *trpRBA* transcript expression by reverse transcription quantitative PCR (RT-qPCR) under iron- or tryptophan-limited conditions. First, we compared expression of the operon during transient (6 h) or prolonged (24 h) starvation of either nutrient.

*Ctr*-infected cells were starved for iron or tryptophan by the addition of the iron chelator 2,2-bipyridyl[22–24] or by replacing complete media with defined media lacking tryptophan[24]. We expected prolonged starvation to induce chlamydial persistence, an aberrant developmental state[18,46]. To verify our treatment conditions, we monitored three aspects of chlamydial development: morphology, genome replication, and transcription of persistence biomarkers. As expected, transient starvation did not grossly alter chlamydial morphology or the expression of the persistence biomarkers *euo* and *omcB* (Fig. 1a–c). We observed a significant decrease in genome replication following brief starvation, but this was very modest when compared to the prolonged starvation protocol (Fig. 1b). Indeed, prolonged persistence-inducing conditions resulted in markedly smaller inclusions with aberrantly enlarged organisms (Fig. 1a), a nearly 2-log reduction in genome replication (Fig. 1b) and the expected increase in *euo* expression with the concomitant decrease in *omcB* expression[47] (Fig. 1c).

We then compared expression of *trpRBA* between our iron- and tryptophan-limited conditions. As expected, transient iron starvation specifically induced the expression of *trpBA*, while *trpR* expression was statistically indistinguishable from mock treatment[23,24] (Fig. 1d). In contrast, transient tryptophan limitation robustly increased the expression of *trpRBA* ~100-fold. However, prolonged starvation revealed that iron limitation is unable to maximally induce the expression of *trpRBA* (Fig. 1d). We examined if this was true of other known iron-regulated genes by assaying the expression of *ahpC*, *devB*, *nrdAB*, and *ytgA*[22–24,48]. These genes all reached equivalent levels of expression following prolonged iron or tryptophan starvation (Supplementary Fig. 1). We reasoned that the unique expression of *trpRBA* was attributable to the dual regulation by YtgR and TrpR. While some of the iron-regulated genes we assayed (e.g.,

*ytgA*) are or may be regulated by YtgR, we cannot discount a stress-independent regulatory mechanism that functions during persistence to upregulate these genes. Nevertheless, their expression stands in stark contrast to that of *trpRBA* under the same conditions, suggesting a unique regulatory schematic for the *trpRBA* operon.

Next, we tested whether a transient, but combined, iron and tryptophan starvation, which would inactivate both YtgR and TrpR, further increased *trpBA* expression relative to tryptophan starvation alone. We observed that the combined stress increased *trpBA* expression only to the levels achieved under the tryptophan-starved condition (Fig. 1e), suggesting that tryptophan limitation itself is sufficient to overcome iron-dependent repression. Relative to either stress alone, this combined transient stress condition did not further reduce chlamydial genome replication or produce more aberrant morphology, nor did it show transcriptional signs of persistence based on *euo* and *omcB* expression (Supplementary Fig. 2a–c). Based on these results, we hypothesized that tryptophan availability regulated the activity of YtgR repression at P$_{trpBA}$.

### YtgR is regulated by tryptophan availability through the upstream WWW motif encoded in the YtgC permease domain

YtgCR is highly enriched for tryptophan codons relative to the chlamydial proteome (2.44% versus 0.95%). Indeed, YtgCR is one of only three ORFs in the chlamydial proteome to encode a WWW motif (Fig. 2a), the other two being CTL0174 and CTL0456, putative NadC-like dicarboxylate symporters. Sequential motifs of tryptophan codons are known to play a role in connecting tryptophan availability to translation and transcription, as in transcriptional attenuation of the *E. coli trp* operon (e.g., the WW motif in TrpL). We therefore hypothesized that the WWW motif of YtgCR may sensitize the expression of YtgR to tryptophan levels.

To test the tryptophan dependency of YtgR expression, we analyzed the abundance of YtgR in lysates from infected cells under tryptophan-replete or -deplete conditions by immunoblot (Fig. 2b). We observed that following extended tryptophan limitation, YtgR expression was markedly reduced, similar to the expression of the tryptophan-rich (1.52%) major outer membrane protein, OmpA (a.k.a MOMP; Supplementary Fig. 3a, b). In contrast, expression of the tryptophan-poor (<0.56%) GroEL homologs (GroEL_1–GroEL_3) was unhindered by tryptophan starvation, represented by the normalized signal intensity of GroEL_2 (0.56%; Supplementary Fig. 3c). Notably, the expression of YtgR and OmpA was rescued by the addition of exogenous indole to the tryptophan-depleted media, implicating tryptophan starvation as the cause of reduced expression.

Under normal conditions, YtgR is expressed at very low levels and therefore difficult to detect in lysates from infected cells until late in infection. We therefore generated *Ctr* transformants that ectopically expressed recombinant C-terminally FLAG-tagged YtgCR from an anhydrotetracycline (aTc)-inducible shuttle vector, pBOMBL. Ectopic expression from these constructs thus monitors both full-length and cleaved YtgR expression. To investigate the involvement of the WWW motif in sensitizing YtgR expression to tryptophan availability, we generated an additional transformant where the WWW motif was substituted for the hydrophobic aromatic amino acids tyrosine-tyrosine-phenylalanine (YYF). We validated these constructs by first inducing expression of YtgR-FLAG in *E. coli*, where we observed robust detection of both the FLAG-tagged YtgR product as well as the YtgR cleavage product (Supplementary Fig. 4a). However, despite efforts to optimize conditions for ideal expression in infected cells, we were unable to detect YtgR-FLAG by

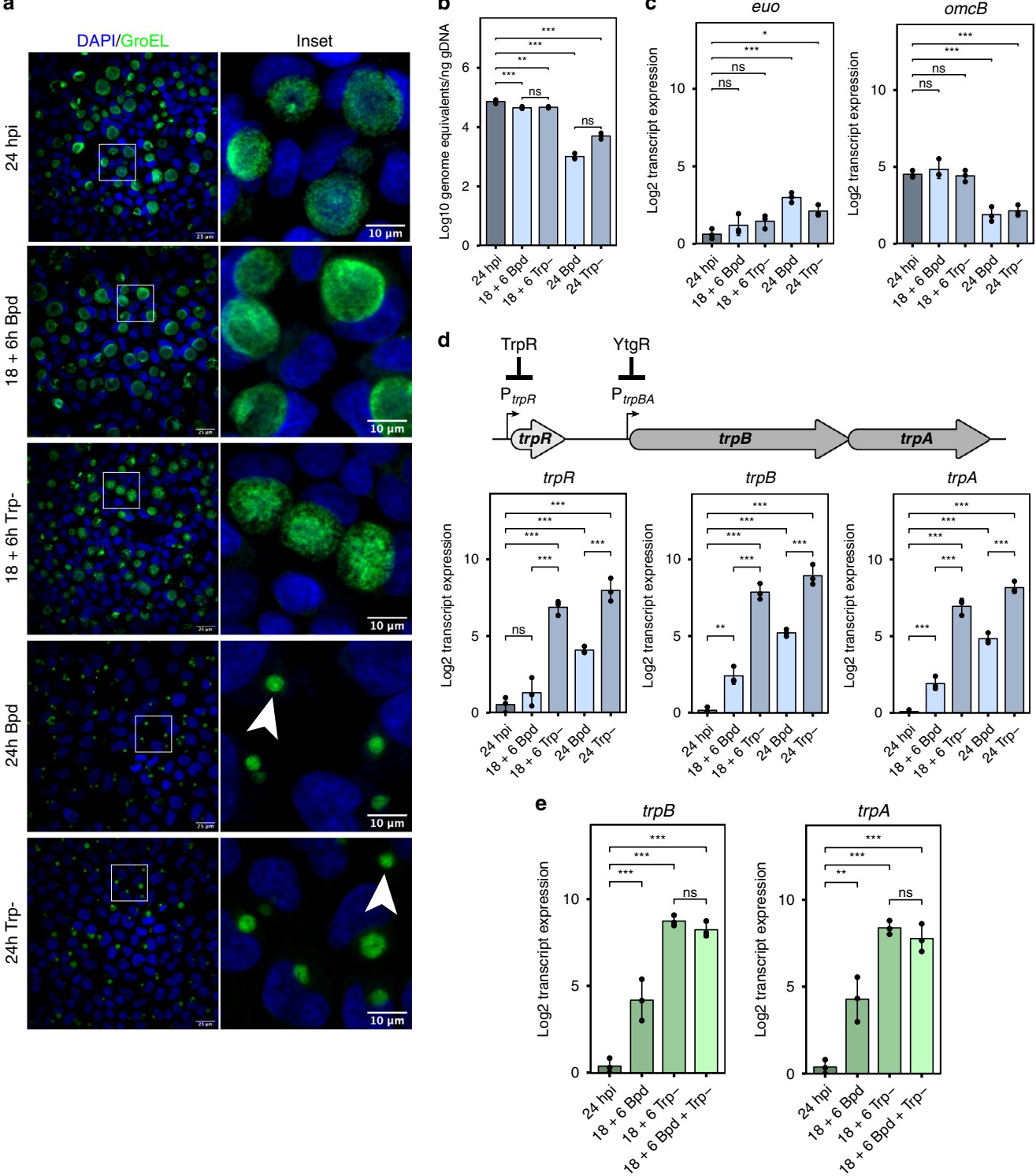

**Fig. 1 Tryptophan limitation overcomes iron-dependent repression of *trpBA*. a** Representative ($N = 3$) immunofluorescent confocal micrographs of chlamydial morphology in HeLa 229 cells infected with *C. trachomatis* serovar L2 (434/Bu) (MOI 2) starved for iron by 100 µM 2,2-bipyridyl treatment (Bpd) or starved for tryptophan by media replacement (Trp-). *C. trachomatis* was visualized by detecting the cytosolic GroEL_1–GroEL_3 antigens. Arrowheads indicate inclusions with visibly enlarged, persistent bacteria. **b** *C. trachomatis* L2 genome copy number following iron or tryptophan starvation determined by quantitative PCR (qPCR) targeting the *euo* open-reading frame. **c** Transcript expression of the persistence biomarkers *euo* and *omcB*. Persistence is generally characterized by an increase in *euo* expression and a decrease in *omcB* expression relative to mock treatment (24 hpi). **d** (Top) Schematic representation of the *trpRBA* operon. The major TrpR-regulated promoter (P*trpR*) and alternative YtgR-regulated promoter (P*trpBA*) are annotated at their respective positions. (Bottom) Transcript expression of the *trpRBA* operon following either iron or tryptophan limitation. **e** Transcript expression of *trpBA* following transient 6 h starvation of iron, tryptophan, or both stresses combined. Data represent mean and s.d. of $N = 3$ (independent biological replicates). Statistical significance in all panels was determined by one-way ANOVA followed by Tukey's post-hoc test of honest significant differences (two-tailed). *$p < 0.05$, **$p < 0.01$, ***$p < 0.001$. Source data for this figure are provided in Source Data.

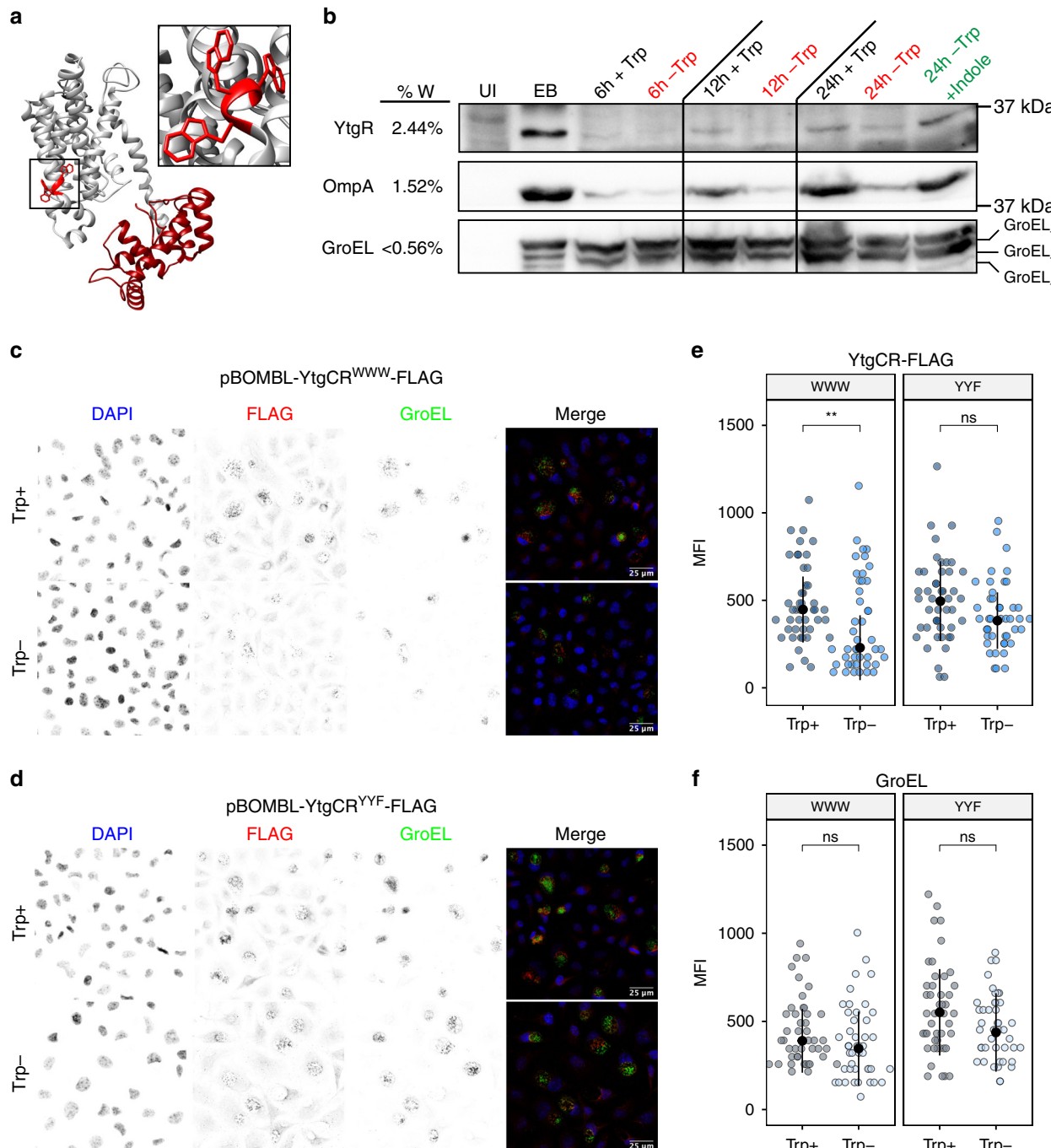

**Fig. 2 Tryptophan-dependent expression of YtgR is mediated by the WWW motif in YtgC. a** Ab initio structural model of YtgCR determined by the Phyre2 server. Inset shows the WWW motif. The gray region is the predicted YtgC ABC-type permease domain. The crimson region is the DtxR-like YtgR repressor domain. **b** Immunoblot of YtgR expression from *C. trachomatis* L2-infected HeLa 229 cells (MOI 5) under tryptophan-replete (Trp + ) or -deplete (Trp-) conditions. Expression of tryptophan-rich OmpA and tryptophan-poor GroEL_1–GroEL_3 were monitored as controls. Tryptophan starvation was rescued by the addition of 10 μM indole to the depleted media. UI uninfected. EB lysate from *C. trachomatis* elementary bodies (inoculum). Equivalent volumes of lysates collected in parallel were added to each well. Image is representative of two independent experiments ($N = 2$). **c** Immunofluorescent confocal micrographs of FLAG expression from *C. trachomatis* L2 transformed with either pBOMBL-YtgCR^WWW-FLAG or **d** pBOMBL-YtgCR^YYF-FLAG under tryptophan-replete or -deplete conditions. McCoy B mouse fibroblasts were infected at an MOI of 5 in the presence of Penicillin G to select for transformants. Anhydrotetracycline was added at 20 nM to induce YtgCR-FLAG expression. GroEL was used as a counterstain to identify inclusions for quantification. Images representative of three independent experiments ($N = 3$). **e** Quantification of mean fluorescent intensity (MFI) from the FLAG channel in panels **c** and **d**. Black dot in dotplots indicates the population median and the error bars represent the median absolute deviation. **f** Quantification of MFI from the GroEL channel in panels **c** and **d** as performed above for panel **e**. Statistical significance in all panels was determined by pairwise two-sided Wilcoxon rank-sum test. *$p < 0.05$, **$p < 0.01$, ***$p < 0.001$. Source data for this figure are provided in Source Data.

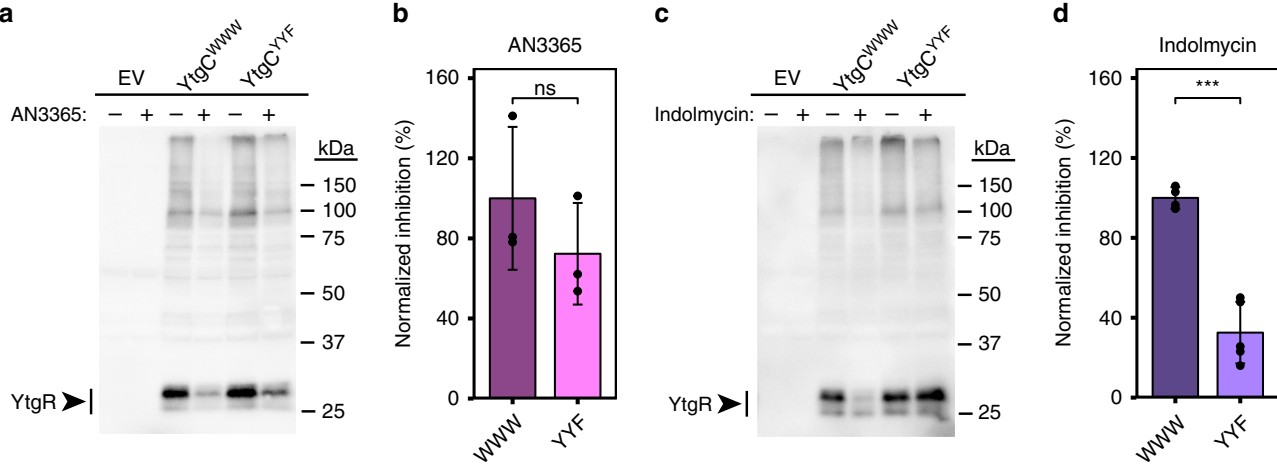

**Fig. 3 Indolmycin inhibits YtgR expression in a WWW motif-dependent manner. a** Immunoblot of pET151-YtgCR-3xFLAG (WWW or YYF) expression from BL21 (DE3) *E. coli* lysates in the presence or absence of 10 μg/mL of the leucyl-tRNA synthetase inhibitor, AN3365. Experiments were performed in M9 minimal media supplemented with 0.2% D-glucose. YtgCR-3xFLAG was detected with a monoclonal FLAG antibody. Images are representative of at least three independent experiments. **b** Densitometric quantification of YtgR expression following AN3365 treatment relative to the mock treated control group and normalized to the mean of the WWW group. **c** Equivalent experiment to that described in **a** with the exception that cultures of transformed *E. coli* were treated with 1.0 μg/mL of the tryptophanyl-tRNA synthetase inhibitor, indolmycin. **d** Equivalent quantification to that described in **b** for indolmycin-treated experiments. Data represent mean ± s.d. of at least three independent biological replicates. Statistical significance in all panels was determined by pairwise two-sided unpaired Welch's *t*-test for unequal variance. *$p < 0.05$, **$p < 0.01$, ***$p < 0.001$. Source data for this figure are provided in Source Data.

immunoblot under any conditions tested (Supplementary Fig. 4b). We additionally generated YtgCR-3xFLAG-tagged constructs, the induced expression of which could be detected by immunoblot (Supplementary Fig. 4c). However, all attempts to inhibit YtgR-3xFLAG induction were inconclusive, suggesting that the extreme induction period required to detect YtgR-3xFLAG precluded our ability to rely on immunoblot to observe tryptophan- and WWW motif-dependent changes in expression (Supplementary Fig. 4c±e). We therefore turned to immunofluorescent confocal microscopy, where we could easily detect specific FLAG signal following periods of brief aTc induction (Supplementary Fig. 4f, g) and with negligible background signal (Supplementary Fig. 4h, i). This also afforded us the ability to better account for inter-inclusion heterogeneity in the expression of inducible YtgCR-FLAG.

We analyzed the expression of YtgCR^WWW and YtgCR^YYF under tryptophan-replete and -deplete conditions to assess the role of the WWW motif in regulating the tryptophan sensitivity of YtgR expression (Fig. 2c, d). Cells infected with each transformant were tryptophan-starved prior to a brief aTc induction period and FLAG signal was quantified from each condition. We observed that tryptophan depletion significantly reduced the induction of YtgCR^WWW expression, while the inducibility of YtgCR^YYF was relatively unaffected (Fig. 2e). This is notable given that the YtgR domain itself is tryptophan-rich and may encode six tryptophan codons based on the observed molecular weight of the cleavage product[45]. Analysis of the signal intensity from the GroEL channel demonstrated that levels of the constitutively expressed, tryptophan-poor GroEL homologs were unaffected by tryptophan depletion for either transformant (Fig. 2f). We therefore concluded that the WWW motif plays a predominant role in sensitizing the expression of YtgR to tryptophan availability.

**Inhibition of tryptophanyl-tRNA charging by indolmycin prevents translation of YtgR in a WWW motif-dependent manner**. We hypothesized that WWW motif-dependent tryptophan sensitivity in YtgCR was a consequence of translating ribosomes unable to incorporate tryptophan residues into the

nascent peptide at the motif. This would occur as a function of depleted pools of charged tryptophanyl-tRNAs. To examine whether ribosome stalling at the WWW motif abrogated translation of YtgR, we designed an expression system for YtgCR in *E. coli*, where we expressed YtgCR^WWW or YtgCR^YYF C-terminally tagged with a 3xFLAG epitope from the IPTG-inducible pET151 plasmid. Establishing this system in *E. coli* allowed us to avoid issues with immunoblot sensitivity that we observed with the chlamydial transformants (Supplementary Fig. 4b). We then took advantage of the competitive inhibitor of prokaryotic tryptophanyl-tRNA synthetase, indolmycin[49,50], which prevents the incorporation of tryptophan during translation and has been shown to promote ribosome stalling at tryptophan codons[51]. As a control, we included a treatment with the translation inhibitor AN3365, which functions as a non-competitive inhibitor of prokaryotic leucyl-tRNA synthetase[52,53]. YtgCR contains 66 leucine residues and therefore the effect of AN3365 on YtgCR expression should be independent of any change in the WWW motif. Importantly, our *E. coli* system allowed us to use relatively brief induction and treatment periods, which reduces the possibility that observed changes in expression would be due to downstream effects of translation inhibition.

We transformed our YtgCR expression vectors into BL21 (DE3) *E. coli* and cultured them in minimal media to prevent tryptophan from outcompeting indolmycin. Concomitantly with the addition of IPTG to induce expression of YtgCR, we added either indolmycin or AN3365 and then analyzed YtgR in the lysates by immunoblot with a monoclonal FLAG antibody (Fig. 3a–c). Following induction, we observed that cleavage of YtgCR yielded one major band at the expected molecular weight of YtgR (30 kDa). When we treated the transformed *E. coli* with AN3365, we observed potent inhibition of YtgCR expression irrespective of changes in the WWW motif (Fig. 3a, b). Induction of YtgCR resulted in a significant decrease in bacterial growth, which was partially rescued by AN3365 treatment, verifying the inhibition of YtgCR expression (Supplementary Fig. 5a). In contrast, indolmycin treatment resulted in significant inhibition of YtgR expression only in the WWW background, while

expression of the YYF variant was relatively unaffected (Fig. 3c, d). Additionally, indolmycin treatment only rescued bacterial growth from the expression of YtgCR$^{WWW}$ and not YtgCR$^{YYF}$ (Supplementary Fig. 5b). Therefore, during periods of tryptophan limitation when charged tryptophanyl-tRNAs are less abundant, ribosomes likely stall at the WWW motif and restrict YtgR expression.

**Indolmycin treatment inhibits transcription of YtgCR 3′ of the WWW motif in a Rho-independent fashion.** In *Chlamydia*, tryptophan limitation influences the transcription of tryptophan codon-rich genes; in response to IFNγ-mediated tryptophan starvation, genes or operons enriched for tryptophan near the 5′-end adopt a polarized expression resulting in depletion of the 3′-end of the transcript[54,55]. One such operon that is susceptible to this regulation is the *ytgABCD* operon, where tryptophan limitation lowers expression of *ytgD* relative to *ytgA* due to a higher abundance of 5′ tryptophan codons (Fig. 4a). Treatment of *Chlamydia*-infected cells with indolmycin also polarizes transcription of the *ytg* operon, mimicking IFNγ-mediated depletion of host cell tryptophan pools[56]. This regulation is mediated by the chlamydial ortholog of the transcription termination factor Rho, whereby ribosome stalling at tryptophan codon-rich regions of a transcript enables binding of Rho to rho utilization (rut) sites to terminate downstream transcription[57]. We therefore sought to interrogate the influence of chlamydial Rho in terminating transcription downstream of the WWW motif within the YtgCR ORF.

To examine whether tryptophan limitation polarized *ytgCR* transcription, we monitored transcript levels 3′ of the WWW motif relative to amplicon levels 5′ of the motif by RT-qPCR in the presence of either indolmycin or IFNγ. We additionally assayed three other genes: *groEL_1*, which contains no tryptophan codons, *ompA*, which is tryptophan-rich but lacks sequential tryptophan codon motifs, and *ctl0174* which is both tryptophan-rich and contains a WWW motif. For these experiments, primer sets were designed to flank the WWW motif in *ytgCR* and *ctl0174*. For *groEL_1* and *ompA*, amplicons were chosen that were similarly spaced to those in *ytgCR* and *ctl0174*. We observed that *ytgCR* experienced a significant 3′ polarization downstream of the WWW motif in the presence of either indolmycin or IFNγ, whereas no other gene demonstrated tryptophan-dependent transcriptional polarity (Fig. 4b). These data suggested that the WWW motif in YtgCR uniquely coordinates translation and transcription to reduce expression of the 3′ end of the ORF. Interestingly, media-defined tryptophan limitation did not reproduce these results, suggesting that this effect is the consequence of severe tryptophan starvation and may depend on the depletion of host tryptophan pools in vivo (Supplementary Fig. 6).

We reasoned that the unique transcriptional polarity of *ytgCR* may be due to Rho-mediated transcription termination, given the previously reported role of Rho in the polarization of the *ytg* operon[57]. We first surveyed the polarization of our set of analytes in the presence of the specific inhibitor of Rho, bicyclomycin[58,59], under normal growth conditions. As expected, we observed that polarization of the *ytg* operon was reduced by Rho-inhibition; similarly, *groEL_1* and *ctl0174* polarization was Rho-regulated (Supplementary Fig. 7). However, polarization of *ytgCR* was unaffected by bicyclomycin, indicating that it is not a likely Rho target. Accordingly, Rho-inhibition was unable to rescue *ytgCR* polarization following indolmycin treatment, despite rescuing *ytg* operon polarization (Fig. 4c). The polarization of genes unaffected by indolmycin or Rho inhibition was not rescued under these conditions. Transcriptional polarity of *ompA* was

induced under these conditions, perhaps owing to the longer duration of indolmycin treatment required to include bicyclomycin treatment. However, *ctl0174* did not experience the same defect, suggesting that polarization is mediated by other features of the transcript than tryptophan codon-richness alone. In total, we concluded from these data that severe tryptophan limitation reduces transcription of the *ytgR* domain 3′ of the WWW motif and that this occurs independently of Rho-mediated termination.

**Tryptophan limitation activates the iron-dependent, YtgR-regulated *trpBA* promoter.** With tryptophan depletion reducing the levels of the YtgR repressor, it follows that the expression of genes repressed by YtgR, such as *trpBA*, should be induced under the same condition. We have previously identified an YtgR-regulated alternative TSS, P$_{trpBA}$, that expresses *trpBA* independently from *trpR*[24]. We therefore sought to understand how reducing YtgR levels in the absence of tryptophan might influence expression from P$_{trpBA}$. Using RT-qPCR and Rapid Amplification of 5′ cDNA Ends (5′-RACE), we investigated transcription initiation events across the *trpRBA* operon under iron- and tryptophan-depleted conditions.

First, we asked whether *trpB* was expressed independently of *trpR* under our validated starvation conditions (see Fig. 1) by quantifying the abundance of *trpB* transcripts (expressed from P$_{trpR}$ and P$_{trpBA}$) relative to *trpR* (expressed only from P$_{trpR}$). These data were expressed as a *trpB*:*trpR* ratio, such that a value greater than 1.0 indicates elevated expression of *trpB* independent from *trpR*. Normal developmental conditions indicate that *trpB* and *trpR* transcripts are expressed at relatively equal levels, indicated by a ratio near 1.0 (Fig. 5a). This could be attributable to complete transcription of the *trpRBA* polycistron originating at P$_{trpR}$, thus expressing equivalent *trpBA* transcripts, or expression of *trpR* from a truncated P$_{trpR}$ transcript and expression of *trpBA* from P$_{trpBA}$ at relatively equal levels. Contrastingly, transient or prolonged limitation of either iron or tryptophan resulted in an increase in the abundance of *trpB* relative to *trpR*. The novel finding that *trpBA* is independently expressed from *trpR* during tryptophan limitation is parsimoniously explained by increased transcription from P$_{trpBA}$ due to YtgR inactivation. Consistent with previous reports[35], we were unable to immunoprecipitate fragments of the *trpBA* promoter region with endogenous TrpR despite substantial enrichment of the *trpR* promoter (Supplementary Fig. 8), thus excluding TrpR as a possible mediator of tryptophan-dependent regulation of P$_{trpBA}$.

We next determined the specific TSS for *trpBA* under iron- or tryptophan-depleted conditions. We amplified 5′-RACE products from *Ctr*-infected HeLa cells and separated them by gel electrophoresis to visualize TSS preference under our validated starvation conditions (Fig. 5b). Treatment with bipyridyl for either a transient or prolonged period yielded a 1.0-kb RACE product that mapped near P$_{trpBA}$. The same RACE product was obtained during brief and prolonged tryptophan depletion, which we attribute to reduced levels of YtgR (see Fig. 2). Unlike bipyridyl treatment, however, tryptophan depletion additionally activated the major P$_{trpR}$ promoter element due to inactivation of TrpR in the absence of the co-repressor, tryptophan. We then isolated the RACE products corresponding to P$_{trpR}$ and P$_{trpBA}$ following 24 hours of iron or tryptophan starvation and sequenced them to identify the TSS indicated by the 5′-most nucleotide from each RACE product. Iron limitation, as previously reported, induced *trpBA* expression from a TSS located at genome position 512,005 (Fig. 5c)[24]. Following tryptophan starvation, we detected *trpR* expression originating at 511,389, in agreement with previous reports of the TSS for *trpR*[24,34,35] (Fig. 5d). Notably, tryptophan limitation induced

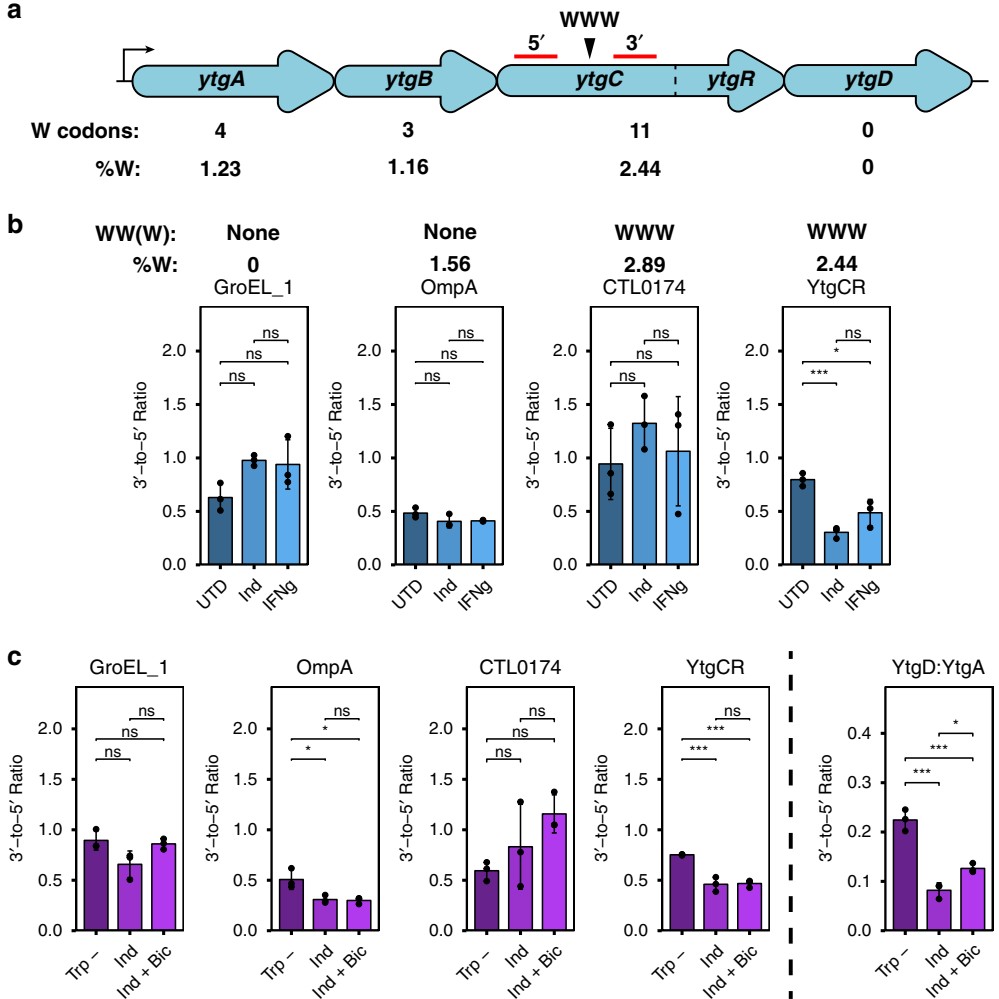

**Fig. 4 Severe tryptophan limitation terminates YtgR transcription in a Rho-independent manner. a** Schematic of the *ytgABCD* operon in *C. trachomatis* L2, with the tryptophan (W) codon content of each open-reading frame indicated below. The location of the WWW motif in the YtgC permease domain is indicated above the operon schematic. Red lines indicate 5′ and 3′ amplicons for RT-qPCR analysis. **b** Analysis of transcriptional polarization by RT-qPCR for YtgCR, GroEL_1, OmpA, and CTL0174. *C. trachomatis* L2-infected HEp-2 cells were treated with either 120 μM indolmycin (Ind) in tryptophan-depleted media, 0.5 ng/mL IFNγ or left untreated (tryptophan-replete media; UTD) and transcriptional polarization was assessed by comparing 3′-to-5′ gene expression. The W codon content and presence of sequential W codon motifs is indicated above the plots for each gene. Note that these samples were prepared for a prior publication[56] and thus the previously reported methodology applies. **c** Assessment of the involvement of the termination factor Rho in mediating transcriptional polarity for the genes described above in **b**. Following indolmycin treatment, *C. trachomatis* L2-infected HeLa cells were treated with 25 μg/mL bicyclomycin (Bic) for 4 h to inhibit Rho activity. Tryptophan-depleted media (Trp-) was used as a reference condition to specifically discern the effect of indolmycin. Transcriptional polarity of the *ytg* operon (YtgD:YtgA) was included as a positive control for rescue by Rho inhibition. Note that YtgD:YtgA is displayed on a different scale due to the more pronounced difference between *ytgA* and *ytgD* expression under normal conditions. Data in all panels represent mean ± s.d. of three independent biological replicates (*N* = 3). Statistical significance in all panels was determined by one-way ANOVA followed by Tukey's post-hoc test of honestly significant differences (two-tailed). *$p < 0.05$, **$p < 0.01$, ***$p < 0.001$. Source data for this figure are provided in Source Data.

*trpBA* expression from the same nucleotide position as iron limitation, 512,005. Together, these data implied that *trpBA* is transcribed from the same YtgR-regulated promoter, $P_{trpBA}$, in response to both iron and tryptophan limitation (Fig. 6).

## Discussion

These data collectively point to a model where tryptophan deprivation limits the level of the YtgR repressor to activate transcription from the alternative YtgR-regulated promoter for *trpBA* (Fig. 6). YtgR translation is regulated by tryptophan through the upstream WWW motif in the YtgC permease domain to reduce availability of the repressor under tryptophan-starved conditions, thereby de-repressing *trpRBA*. This regulatory

mechanism parallels *cis*-attenuation by TrpL in *E. coli*, though notable differences exist: namely, this attenuation mechanism is mediated by a *trans*-acting transcription factor. While we have focused here on the comparison of YtgCR to TrpL, other mechanisms of *trp* operon attenuation exist. *Bacillus subtilis*, which lacks a TrpR orthologue, has a comparable but distinct attenuation mechanism, where the oligomeric RNA-binding TRAP protein functions in *trans* to sequester the leader RNA of the *trp* operon in a tryptophan-dependent manner[60], thereby attenuating expression of the structural genes of the operon. TRAP-mediated attenuation functions with the same regulatory logic as YtgCR and TrpL, yet through a highly specialized mechanism that does not closely parallel either. It is interesting to note, however, that both TRAP and YtgCR function in *trans*,

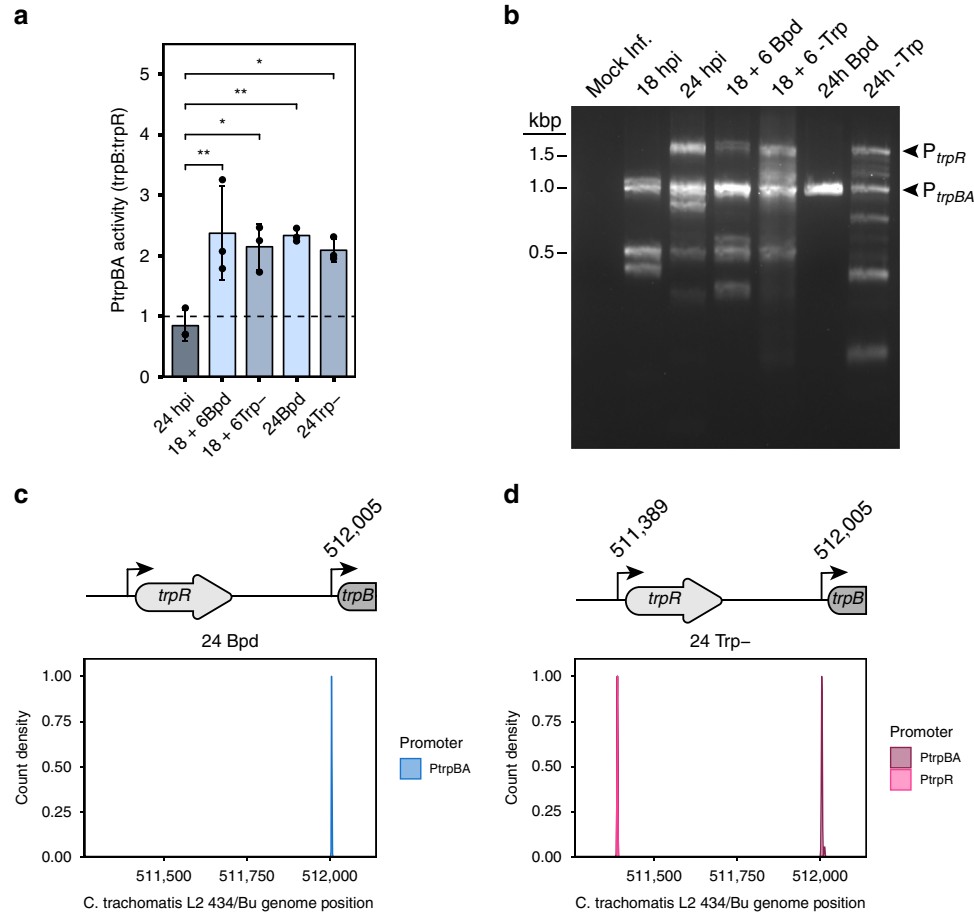

**Fig. 5 Tryptophan starvation activates the YtgR-regulated, iron-dependent promoter for *trpBA*. a** Assay for $P_{trpBA}$ promoter activity determined by RT-qPCR comparison of *trpB*-to-*trpR* transcript levels. Values exceeding 1.0 indicate independent expression of *trpBA* from the alternative promoter. **b** Representative image of 5′-RACE products separated on a 2% agarose gel. The $P_{trpR}$ and $P_{trpBA}$ products are annotated on the right of the gel image. Note that the various apparent products below 1.0 kb were inconsistent across replicates and therefore treated as experimental artefacts. Image is representative of three independent experiments ($N = 3$). **c** Density plot of the 5′-most nucleotide of RACE products extracted from the 24 h Bpd condition. A schematic of the relevant region of the *trpRBA* operon is depicted above the plot with the nucleotide position from the sequence population annotated above the $P_{trpBA}$ transcriptional start site. **d** same as in **c** but for the RACE products isolated from the 24 h -Trp condition for both $P_{trpR}$ and $P_{trpBA}$. At least 15 isolated clones were sequenced across three replicates. Data in barplots represent mean ± s.d. of three independent biological replicates ($N = 3$). Statistical significance was determined by one-way ANOVA followed by Tukey's post-hoc test of honestly significant differences (two-tailed). *$p < 0.05$, **$p < 0.01$, ***$p < 0.001$. Source data for this figure are provided in Source Data.

suggesting the convergent evolution of similar logics for *trp* operon attenuation in highly dissimilar organisms.

An outstanding question is the role of possible *cis*-attenuation in the *trpRBA* operon. A putative *trpL* leader peptide (TrpL$^{Ctr}$) has been annotated in the *trpRBA* IGR in the *Ctr* genome[61,62], but it has not been experimentally validated. TrpL$^{Ctr}$ was inferred from the prediction of alternating RNA hairpin structures upstream of *trpBA*[61], but has many atypical characteristics. Unlike putative TrpL peptides in other bacteria, TrpL$^{Ctr}$ is annotated as an unusually long 59 amino acid peptide with non-consecutive tryptophan codons, thus lacking a signature sequential tryptophan codon motif (Supplementary Data 1). In silico analysis of the IGR by the PASIFIC algorithm[63] indicates that this sequence would form a poor attenuator and it predicts transcript termination before the end of the annotated leader peptide ORF (Supplementary Fig. 9). We suggest the possibility that the evolutionary gain of the YtgR operator sequence in the *trpRBA* IGR may have led to the degeneration of TrpL function as YtgR was able to maintain tryptophan-dependent regulation of *trpRBA*. Future studies will be aimed at elucidating the true function, if any, of the putative *trpL* in *Ctr*.

Somewhat unexpectedly, we report here that the polarized transcription of the *ytgCR* ORF, likely mediated by the WWW motif under tryptophan-starved conditions, is insensitive to Rho inhibition (Fig. 4c; Supplementary Fig. 7). However, the ORF immediately downstream – *ytgD* – is regulated by Rho-dependent transcription termination[56,57], which we have confirmed. Therefore, it appears that two independent mechanisms of transcription termination distinctly regulate *ytgCR* and *ytgD*, despite their co-expression within a polycistronic transcript. We speculate that this regulation depends on the location of the translating ribosomes relative to the WWW motif and the RNA polymerase(s) transcribing the *ytgABCD* operon. As transcription reads-through into the *ytgCR* ORF, the opportunity is presented for tryptophan limitation to stall ribosomes at the WWW motif, thus enabling Rho-independent termination of *ytgCR*. However, for the proportion of transcription events that overcome this hurdle, they may then be subjected to Rho-dependent termination near or within the *ytgD* ORF. This regulatory model of sequential termination events offers a possible explanation for the observation that under normal, steady-state conditions, *ytgD* is expressed at least 5-fold less than *ytgA* (Fig. 4c and Supplementary Fig. 7).

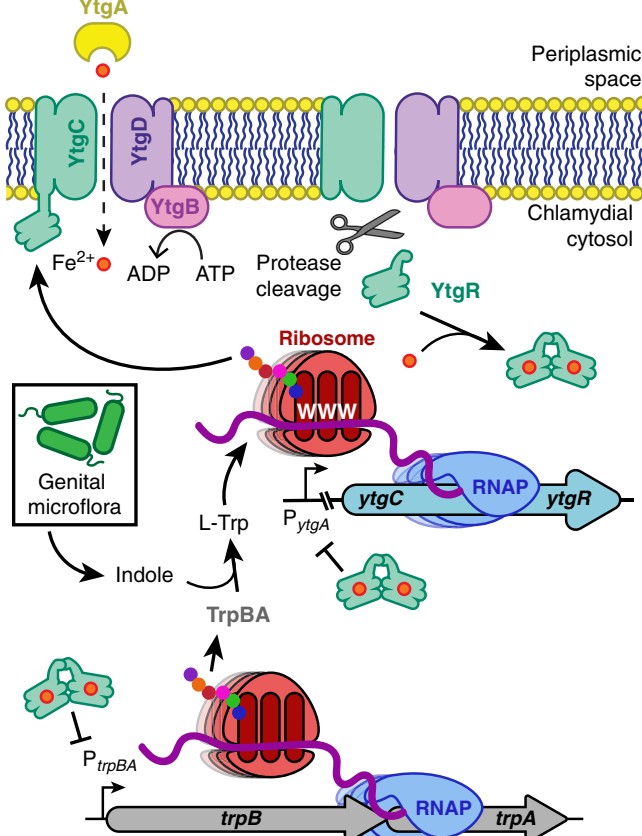

**Fig. 6 Model of tryptophan-dependent regulation of YtgR expression and YtgR-mediated *trpBA* repression.** Under tryptophan-replete conditions, ribosomes are able to read through the WWW codon motif on the nascent *ytgCR* message transcribed by the RNA polymerase (RNAP) to translate the full-length YtgCR. Cleavage from YtgC allows YtgR to dimerize and coordinate iron to gain repressor functionality. YtgR then maintains repression of *trpBA* from the intergenic region. In contrast, tryptophan depletion inhibits ribosome readthrough at the WWW motif, reducing protein expression of YtgR and terminating downstream transcription of the YtgR domain in a Rho-independent manner. Thus, YtgR levels are reduced and *trpBA* is de-repressed by YtgR. This allows for tryptophan salvage from indole provided by the genital microflora via the activity of the TrpBA synthase, which can restore YtgR levels and transcriptional repression of *trpBA*. Some elements in this figure have been used in previously published figures[24,38] and were adapted for use here, with some changes, under the Creative Commons license CC BY 4.0: https://creativecommons.org/licenses/by/4.0/.

Our data indicate that YtgR is a nexus of the chlamydial response to iron and tryptophan limitation. We suspect that the WWW motif performs an important regulatory function during *Ctr* infection of the female LGT by connecting the availability of tryptophan to the availability of iron. As discussed above, the LGT is both indole-rich and iron-poor, which provides necessary context for the utility of iron-dependent YtgR regulation of *trpBA*. The WWW motif then completes the regulatory feedback loop by subjecting the expression of YtgR to tryptophan-dependent regulation. In the absence of iron, *trpBA* can be expressed via YtgR de-repression to salvage tryptophan[24]. Alternatively, if iron is abundant, but tryptophan is absent, reduced YtgR expression will de-repress *trpBA* to salvage tryptophan from indole. Biosynthesis of tryptophan thus permits the full translation of YtgCR to re-establish YtgR repression of *trpBA*. The absence of indole (and by extension, tryptophan) would reduce

YtgR levels via the WWW motif, to the detriment of the pathogen. A recent report described the potential for chlamydial TrpBA to generate toxic levels of ammonia in an off-target reaction that takes place in the absence of indole[64]. The inability to maintain YtgR repression of *trpBA* under these conditions may exacerbate ammonia toxicity to *Ctr*. Tryptophan-dependent regulation by the WWW motif may also impact iron import by the YtgCD permease, thereby affecting iron-dependent YtgR repression. Presently, it is unclear if the YtgC permease domain directly participates in iron acquisition, or if YtgD alone is a sufficient permease for iron import. Nevertheless, it will be interesting to investigate how tryptophan limitation influences iron acquisition by *Ctr*.

A final interesting consideration is the broader role of YtgR regulation in chlamydial development. As a biosensor for both tryptophan and iron levels, YtgR may be critical to maintenance of normal gene expression throughout the chlamydial developmental cycle as it siphons nutrients from the host cell. The WWW motif of YtgCR is highly conserved across the *Chlamydiaceae* but poorly conserved among the environmental chlamydiae in the *Chlamydiales* order (Supplementary Fig. 10a). In contrast, the WWW motif of CTL0174 is perfectly conserved across the *Chlamydiales*, including in *Waddlia chondrophila* and *Parachlamydia acanthamoebae* (Supplementary Fig. 10b). This suggests that the YtgCR WWW motif may have been gained in adaptation to mammalian hosts. This is further supported by the observation that among the four avian chlamydiae included in our analysis, only two contain complete WWW motifs in YtgCR. Discerning the regulatory function of YtgR among chlamydial species may reveal important insights into the evolutionary trajectory of the *Chlamydiales* and the adaptation of *Ctr* to life as a human pathogen.

## Methods

**Eukaryotic cell culture and chlamydial infections**. Human cervical epithelial adenocarcinoma HeLa cells were cultured at 37 °C with 5% atmospheric CO2 in Dulbecco's Modified Eagle Medium (DMEM; Gibco, Thermo Fisher Scientific, Waltham, MA, USA)) supplemented with 10 μg/mL gentamicin, 2 mM L-glutamine, and 10% (v/v) filter-sterilized fetal bovine serum (FBS). For all experiments, HeLa cells were cultured between passage numbers 3 and 15. McCoy B mouse fibroblasts (originally from Dr. Harlan Caldwell, NIH/NIAID) were cultured under identical conditions. *Chlamydia trachomatis* serovar L2 (434/Bu) was originally obtained from Dr. Ted Hackstadt (Rocky Mountain National Laboratory, NIAID). Chlamydial EBs were isolated from infected HeLa cells at 36–40 h post infection (hpi) and purified by density-gradient centrifugation essentially as described[65].

For infections, cells were first washed with Hanks Buffered Saline Solution (HBSS; Gibco, Thermo Fisher Scientific) and ice-cold inoculum prepared in HBSS at the indicated multiplicity of infection was overlaid onto the cell monolayer. To synchronize the infection, inoculated cells were then centrifuged for 15 min at 500xRCF, 4 °C in an Eppendorf 5810 R tabletop centrifuge with an A-4-81 rotor. The inoculum was then aspirated and pre-warmed DMEM (or relevant media with treatment supplementation) was added to the cells. Infected cultures were then returned to the tissue culture incubator until the indicated time post infection.

**Treatment conditions**. For iron starvation and media-defined tryptophan starvation, treatment was performed essentially as described previously[24]. In brief, 100 mM 2,2-bipyridyl (Sigma Aldrich, St. Louis, MO, USA; CAS: 366-18-7) prepared in dimethyl sulfoxide (DMSO) was added to complete DMEM (or tryptophan-depleted DMEM-F12, as described below) at a working concentration of 100 μM at the start of infection (24 h) or at 18 hpi (18 + 6 h). When added after the time of infection, cells were first washed with HBSS prior to bipyridyl treatment. Tryptophan depletion was performed by first washing cells with HBSS and then replacing complete DMEM with tryptophan-depleted DMEM-F12 (U.S. Biological Life Sciences, Salem, MA, USA). Media was replaced either at the time of infection (24 h) or at 18 hpi (18 + 6 h). For immunoblotting, tryptophan starvation began at 12 hpi and samples were collected at the indicated time-points. Indole supplementation was performed by adding 10 μM indole (prepared as 10 mM stock in 100% ethanol) to the tryptophan-depleted media at the time of replacement. For analysis of YtgCR-FLAG expression from transformed *C. trachomatis* strains, 16 h tryptophan starvation began at 8 hpi. Tryptophan-depleted media was supplemented as described for DMEM with the exception that the FBS was dialyzed in a

10 kDa MWCO dialysis cassette (Thermo Fisher Scientific) for 16–20 h at 4 °C in 4 L of PBS. Treated cells were then returned to the tissue culture incubator for the remainder of the experimental time course.

For cell culture experiments, indolmycin treatment was performed essentially as described previously[56]. In brief, 120 mM indolmycin (Cayman Chemical, Ann Arbor, MI, USA; CAS: 21200-24-8) prepared in DMSO at a working concentration of 120 µM at 8 hpi and proceeded for 20 h.

Bicyclomycin treatments were performed essentially as described previously[57]. In brief, 25 mg/mL bicyclomycin benzoate (Cayman Chemical; CAS: 37134-40-0) prepared in DMSO was added to infected cell cultures at a working concentration of 25 µg/mL at 24 hpi (16 h after addition of indolmycin) and treatment was allowed to proceed for 4 h. For the initial survey of Rho-dependent regulation among the assayed genes, bicyclomycin was added at 18 hpi and treatment was allowed to proceed for 6 h.

**Nucleic acid preparation**. RNA was harvested from *C. trachomatis*-infected cells by scraping one or two wells of a 6-well tissue culture plate in a total volume of 500 µL Trizol Reagent (Thermo Fisher Scientific). Samples were transferred to RNase-free o-ring capped tubes containing ~100 µL volume of zirconia beads and thoroughly vortexed for 10 min to rupture bacterial cells. Zirconia beads were pelleted by centrifugation at 21,000 × g for 10 min at 4 °C and supernatant was transferred to an RNase-free tube containing 100 µL chloroform (Sigma Aldrich). Samples were vortexed for 15 s prior to a 10 min RT °C incubation. Phases were then separated by centrifugation at 21,000 × g for 15 min at 4 °C. The aqueous top layer was transferred to an RNase-free tube containing 250 µL 100% ethanol to precipitate RNA. Samples were briefly vortexed and then applied to an RNA collection column provided in the PureLink™ RNA Mini Kit (Invitrogen, Thermo Fisher Scientific). RNA was isolated as described by the manufacturer with an on-column DNA digestion using the PureLink™ DNase Set (Invitrogen, Thermo Fisher Scientific). RNA was eluted in nuclease-free H₂O and stored at −20 °C for short-term storage or −80 °C for long-term storage.

Complementary DNA (cDNA) was generated using 1–2 µg of RNA as a template for the SuperScript IV Reverse Transcriptase (RT) VILO master mix (Invitrogen, Thermo Fisher Scientific) with a no-RT control reaction in a half-reaction volume following manufacturer protocols. The no-RT control sample was screened for DNA contamination by end-point PCR targeting the *C. trachomatis* L2 *ompA* ORF. For RACE experiments, cDNA was generated following the SMARTer® 5'/3' RACE kit protocol (Takara Bio, Kusatsu, Shiga Prefecture, Japan).

Genomic DNA (gDNA) was harvested from parallel well(s) of a 6-well plate in 200 µL ice-cold PBS + 10% Proteinase K and processed through the DNeasy Blood and Tissue Kit following manufacture protocols (QIAGEN, Hilden, Germany). gDNA was stored at −20 °C for short-term storage or −80 °C for long-term storage.

**Quantitative PCR**. All quantitative PCR (qPCR) assays were performed using Power Up™ SYBR™ Green Master Mix (Applied Biosystems, Thermo Fisher Scientific) essentially as previously described[24]. In brief, cDNA was diluted 1:5–1:10 and gDNA was diluted 1:50–1:100 in nuclease-free H₂O (dilutions were identical within each experiment). The 2X PCR master mix was diluted to 1X in nuclease-free H₂O with specific primers diluted to 500 nM (see Supplementary Data 2 for complete list of primers). To 79 µL of the master mix solution, 3.3 µL of template (cDNA or gDNA) was added and then aliquoted into three 25 µL technical replicate reactions in a 96-well optical plate. Reactions were analyzed on an Applied Biosystems 7300 Real Time PCR System or a QuantStudio™ 3 Real-Time PCR System with standard SYBR cycling conditions. All assays were performed with a melt-curve analysis to ensure specific product amplification across samples. All primer sets used in qPCR were validated against a standard curve of *C. trachomatis* L2 gDNA diluted from $2 \times 10^{-3}$ to $2 \times 10^{0}$ ng per reaction. $C_t$ values generated from each experimental reaction were then fit to a standard curve and only primer sets with an efficiency of 100 ± 5% were used. All primer pairs were validated for specific product amplification by melt-curve analysis and gel electrophoresis of amplified products.

Genome equivalents (GE) were calculated by first converting the mean $C_t$ of the triplicate technical replicate reactions to a ng quantity of gDNA (ng template) with the linear equation generated from the standard curve of the *euo* primer pair. This value was then normalized to the total ng/µL gDNA isolated for each sample as follows:

$$\text{GE} = \frac{\text{ng template}}{\frac{\text{ng}}{\mu L} \text{ gDNA}}$$

Transcript expression (TE) normalized to genome equivalents was calculated as follows:

$$\text{TE} = \frac{2^{(C_{t_{GE}} - C_{t_{TE}})}}{\frac{\text{ng}}{\mu L} \text{gDNA}}$$

where $C_t$(GE) is the $C_t$ value of genome equivalents from the respective sample, $C_t$(TE) is the $C_t$ value from the cDNA for the same sample and ng/µL gDNA is the concentration of gDNA from that sample. These values were then scaled such that the lowest value across the biological replicates equaled 1.0 for esthetic purposes. The scaled values were then Log2-transformed to enable the statistical comparison

of values that varied widely in magnitude (e.g., *trpRBA* expression following tryptophan starvation vs. bipyridyl treatment). When different dilution factors were used between replicates, $C_t$(GE) and $C_t$(TE) were corrected for dilution.

Calculation of 3'-to-5' ratio and the *trpB:trpR* ratio was performed as follows:

$$\text{Ratio} = 2^{C_t(\text{Ref}) - C_t(\text{Exp})}$$

where $C_t$(Ref) is the $C_t$ value of the reference amplicon (e.g., 5' or *trpR*) and $C_t$(Exp) is the $C_t$ value of the experimental amplicon (e.g., 3' or *trpB*).

**Immunofluorescent confocal microscopy**. To analyze inclusion morphology, HeLa cells were seeded onto acid-washed glass coverslips in 24-well tissue culture plates and infected at MOI = 2. At the indicated times post infection, coverslips were washed with phosphate-buffered saline (PBS) and cells were fixed with 4% paraformaldehyde in PBS for 15 min at RT °C. Fixation solution was then aspirated and coverslips were either stored at 4 °C in PBS or immediately processed for immunofluorescence assays by permeabilizing cells in PBS + 0.2% Triton X-100 (Thermo Scientific™) for 10 min at RT °C with rocking. Permeabilization solution was then decanted and coverslips were washed 3x with PBS. Coverslips were blocked in PBS + 3% bovine serum albumin (BSA) for 1 hr at RT °C with rocking. Coverslips were then washed 3x with PBS prior to being overturned on a 50 µL droplet of PBS + 3% BSA containing primary antibody diluted 1:1000. To detect chlamydial GroEL, cells were stained with monoclonal mouse anti-cHsp60 (MA3-023, Invitrogen, ThermoFisher Scientific). Coverslips were incubated on primary antibody solution overnight at 4 °C in an opaque humidified container. Coverslips were then washed thoroughly by repeated submersion (~50x) in 100 mL PBS before being overturned on a 50 µL droplet of PBS + 3% BSA + 1:1000 secondary antibody + 2.5 µg/mL 4',6-diamidino-2-phenylindole (DAPI) to label nuclei. A donkey anti-mouse AlexaFluor-488 secondary antibody (Invitrogen, Thermo Fisher Scientific) was used to label the primary mouse anti-cHsp60. Coverslips were then incubated for at least 1 h at RT °C in an opaque humidified container prior to being washed as described above in Milli-Q H₂O and then being mounted on glass microscope slides with 10 µL Shandon™ Immu-Mount (Thermo Fisher Scientific). Mounting medium was allowed to solidify overnight. Confocal microscopy was performed on a Nikon Ti2 Eclipse spinning-disk confocal microscope. All images were acquired using identical laser power and exposure settings. To enhance visualization of inclusion morphology, contrast and brightness were adjusted as necessary for each condition in Fiji ImageJ[66]. All images are summed Z-projections of Z-stacks spanning the entire depth of the inclusions in the field.

**Structural modeling of YtgCR**. The amino acid sequence of CTL0325 (YtgCR) from the *C. trachomatis* L2 434/Bu genome (NCBI Accession: NC_010287) was submitted to the Phyre2 server using "Intensive" modeling mode[67]. The resultant. pdb file was then visualized on the UCSF Chimera software and edited as shown[68].

**Immunoblotting**. Lysates of infected HeLa cells were prepared by scraping three wells of a 6-well plate into 300–500 µL volume of ice-cold Mammalian Protein Extraction Reagent (M-PER™) buffer (Thermo Fisher Scientific) supplemented with cOmplete™ Mini EDTA-free Protease Inhibitor Cocktail (Roche, Basel, Switzerland). Scraped cells were allowed to incubate on ice in lysis buffer for 30 min prior to the addition of 1X concentration of Laemmli SDS buffer. Samples were then boiled at 95 °C for 10 min prior to centrifugation at 21,000 × g for 10 min, 4 °C to pellet debris. Lysates were stored at −80° C.

For validation of pBOMBL-YtgCR-FLAG constructs, lysates were collected under CPAF-inhibiting conditions by lysing in boiling hot 1% SDS buffer[69] and then processed to analyze FLAG and GroEL expression as below.

Equal volumes of lysate were then separated by gel electrophoresis on a 12% acrylamide gel in Tris-Glycine-SDS buffer for 1.5 h at 150 V (constant voltage) on a Bio-Rad PowerPac™ HC High-Current Power Supply. Blotting to 0.45 µm nitrocellulose membrane was performed on a Bio-Rad Trans-Blot® SD Semi-Dry Transfer Cell with Bjerrum and Schafer-Nielsen transfer buffer (48 mM Tris, 39 mM glycine, 20% Methanol (v/v)) at 20 V for 60–70 min at RT °C. The membrane was then equilibrated in Tris-buffered saline + 0.1% Tween-20 (TBS-T) prior to blocking in TBS-T + 5% non-fat dry milk (NFDM) for 1 h at RT °C with rocking. Membrane was then washed with TBS-T and TBS-T + 5% NFDM + 1:500–1:1000 primary antibody was added. For detecting YtgR, a polyclonal rabbit antibody was raised against the final 49 C-terminal residues of the YtgCR ORF (Li International, Denver, CO, USA) and used at a 1:500 dilution. OmpA was detected with a monoclonal mouse antibody (clone L2-1-45; a generous gift of Dr. Harlan Caldwell) and used at a 1:1000 dilution. GroEL was detected by monoclonal mouse antibody that detects antigens present on all three GroEL paralogs (MA3-023, Invitrogen, ThermoFisher Scientific) and used at a 1:1000 or 1:5000 dilution. Primary antibody incubation was allowed to proceed overnight at 4 °C with rocking. Primary antibody solution was then washed out 3× with TBS-T and TBS-T + 5% NFDM + 1:2000 Dako goat anti-rabbit or rabbit anti-mouse HRP-conjugated immunoglobulins (Agilent, Santa Clara, CA, USA) was added. Secondary antibody incubation was allowed to proceed for at least 1 h at RT °C with rocking. Secondary antibody solution was then washed out 3× with TBS-T and the membrane was exposed to Immobilon Western Chemiluminescent HRP Substrate (EMD

Millipore Sigma) for ~60 s prior to imaging on a Bio-Rad ChemiDoc XRS + using detection settings appropriate for the band intensity of each analyte.

Band intensity of YtgR, OmpA, and GroEL_2 analytes were quantified by densitometry using Fiji ImageJ. Peak areas were determined for the GroEL_1 reference band for normalization, as well as YtgR and OmpA. Due to high background signal, YtgR peak area was determined from the baseline of the background signal as opposed to the baseline of the total signal.

**Chlamydial transformation.** Transformation of plasmidless *C. trachomatis* serovar L2 was performed essentially as previously described[57]. In brief, 2 µg of deme-thylated plasmid was used to transform $10^6$ EBs in Tris/CaCl$_2$ buffer (10 mM Tris, 50 mM CaCl$_2$, pH 7.4) for 30 min at RT ˚C. The transformation inoculum was then used to infect a confluent monolayer of McCoy B mouse fibroblasts by cen-trifugation at $400 \times g$ for 15 min followed by incubation at 37 ˚C for 15 min. The inoculum was then aspirated, and complete DMEM was added to the well. At 8 hpi, the medium was replaced with DMEM containing 1 unit/mL penicillin G. Infected cells were passaged at 48 hpi and 48 h thereafter until penicillin-resistant bacteria were isolated. Transformants were then expanded and infectious progeny purified by density-gradient centrifugation. Plasmid DNA was isolated from infected cells and transformed into chemically competent *E. coli* for plasmid isolation and subsequent sequencing to verify each transformant.

**Analysis of YtgCR-FLAG expression.** Induction of YtgCR-FLAG expression was performed by the addition of 20 nM anhydrotetracycline (aTc; prepared as 20 µM stock solution in sterile-filtered diH$_2$O) and allowed to proceed for the indicated times. Immunofluorescent confocal micrographs of YtgCR-FLAG expression from *C. trachomatis* L2 transformants were acquired on a Zeiss LSM800 Airyscan point-scanning confocal microscope using identical exposure and laser power setting between samples and conditions on a 40× oil objective. All immunostaining was performed using the protocol described above for the indicated antigens herein. Fields were chosen based on the reference GroEL-488 channel (detected with the same antibody as described above). FLAG expression was detected using a monoclonal rabbit anti-FLAG antibody (Cell Signaling Technology, #14793 S) and a goat anti-rabbit AlexaFluor-594 secondary antibody (Thermo Fisher Scientific). Z-stacks spanning the entire depth of the inclusions in the field were acquired in 0.3 µm slices. Summed Z-projections were then produced from the Z-stacks in Fiji ImageJ (version 2.0.0-rc-69/1.52p, build 269a0ad53f). From the GroEL-488 chan-nel, a threshold mask was created using the Triangle method and default settings. The Analyze Particles tool was then used to define regions of interest (ROI) with a minimum size of 30 µm. This necessarily excluded some inclusions that lacked contiguous or robust GroEL staining. Thereby, inclusion selection was randomized. The ROIs were then applied to the FLAG-594 channel and the Measure tool was used to analyze mean fluorescent intensity per pixel from the defined regions. Five fields in total were measured for each condition per replicate (total of 15 fields). From these measurements, three values were selected randomly per field (45 measurements total) and further analyzed for statistically distinguishable differences. GroEL signal quantification was performed exactly as described above for the GroEL-488 channel on the same inclusion areas defined by the GroEL mask.

**Expression of YtgCR in *E. coli*.** Chemically competent BL21 (DE3) *E. coli* (Sigma Aldrich) were transformed with pET151-EV, -YtgCR$^{WWW}$-3xFLAG or -YtgCR$^{YYF}$-3xFLAG and selected on lysogeny broth (LB) agar plates containing 50 µg/mL carbenicillin (Carb). Resistant colonies were selected and cultured overnight in LB + 50 µg/mL Carb. The following morning, optical density was measured at 600 nm (OD$_{600}$) and transformants were subcultured at 0.4 OD$_{600}$ units into M9 minimal medium (prepared from 5× M9 Minimal Salts [Sigma Aldrich]) supplemented with 2 mM MgSO$_4$, 100 µM CaCl$_2$, 0.2% D-glucose, and 50 µg/mL Carb. Cultures were incubated at 37 ˚C with 300 RPM shaking for 2 h prior to addition of 500 µM isopropyl β-d-1-thiogalactopyranoside (IPTG) with either vehicle (DMSO), indol-mycin, or AN3365. Indolmycin was prepared at 1.0 mg/mL in DMSO and used at a working concentration of 1.0 µg/mL. AN3365 (Cayman Chemical; CAS: 1234563-16-6) was prepared at 1.0 mg/mL in DMSO and used at a working concentration of 10 µg/mL. Incubation proceeded for another 2 h prior to measurement of final OD$_{600}$ and collection of cell pellets for lysate preparation. Lysates were prepared by resuspending the bacterial cell pellet in 150 µL Complete Bacterial Protein Extrac-tion Reagent (B-PER; Thermo Fisher Scientific) supplemented with cOmplete, Mini, EDTA-free protease inhibitor cocktail (Roche). Lysates were incubated at RT ˚C for 15 min prior to the addition of 1× SDS-Laemmli buffer and boiling at 95 ˚C for 10 min. Cell debris was pelleted by centrifugation and samples were processed for immunoblotting as described above using equivalent OD amounts for each sample. FLAG expression was detected using a monoclonal mouse anti-FLAG antibody (Cell Signaling Technology, Danvers, MA, USA; #8146 S). Quantification of band intensity was performed using Fiji ImageJ.

For pBOMBL-YtgCR-FLAG validation, cultures containing 50 µg/mL Carb were inoculated with fresh BL21(DE3) transformants of either the WWW or YYF variant and incubated for 6 h prior to storage at 4 ˚C overnight. The following morning, sub-cultures were inoculated at an OD of 0.1 and allowed to grow for 1 h prior to induction with 100 nM aTc for 3 h. Lysates were then collected as described above

and 0.2 OD units were analyzed by immunoblot using either the monoclonal mouse FLAG antibody or the polyclonal rabbit YtgCR antibody, as above.

**5′-rapid amplification of cDNA ends (5′-RACE).** 5′-RACE was performed essentially as described previously[24]. In brief, 250 ng of RNA isolated from each condition was converted into cDNA and processed through the SMARTer® 5′/3′ RACE Kit (Takara Bio) for 5′-RACE following manufacturer protocols. Half-reaction volumes were used at each step. Gene-specific primers designed to amplify in the 3′-to-5′ direction from within the *trpB* ORF were used for the primary RACE amplification and then a nested primer was used for the secondary RACE amplification to generate specific RACE products (see Supplementary Data 2 for a complete list of primers used in this study). RACE products were electrophoresed on a 2% agarose gel and visualized on a Bio-Rad ChemiDoc XRS + gel imager. RACE products were then gel extracted using the NucleoSpin® Gel and PCR Clean-Up kit following manufacturer protocols. Gel extracted RACE products were cloned into the manufacturer-supplied pRACE sequencing vector using the In-Fusion® homology-directed (HD) Cloning Plus kit following manufacturer pro-tocols. In-Fusion cloning reactions were then transformed directly into Stellar™ chemically competent *E. coli* and selected for on LB agar plates containing 50 µg/mL carbenicillin. Colonies were then screened for inserts using M13 forward and reverse primers to amplify the insert region in its entirety. Colony PCRs were performed using the high-fidelity Platinum™ SuperFi™ DNA polymerase (Invitro-gen, Thermo Fisher Scientific). PCR products were then isolated using the NucleoSpin® Gel and PCR Clean-Up kit. Isolated PCR products were sent for third-party sequencing at Eurofins Genomics (Louisville, KY, USA) with the M13 Forward primer to sequence into the gene-specific end. Sequencing results were then BLASTed against the *C. trachomatis* L2 434/Bu genome (NCBI Accession: NC_010287.1) and the 5′-most nucleotide was recorded and plotted as a density plot. For the 24 Bpd condition and the 24 Trp-P$_{trpR}$ samples, all sequences aligned to the same 5′ nucleotide position, therefore two "FALSE" data points were added flanking each side of the reported value (e.g., 512,004 and 512,006 for 24 Bpd) to create a density around the reported nucleotide position. Sequencing data can be found in Supplementary Data 3. BLAST results are reported in Supplementary Data 4. Source data and source code can be found in Source Data and Supple-mentary Software, respectively.

**Chromatin immunoprecipitation of *Chlamydia*-infected cells.** Confluent monolayers of HeLa cells from six-well tissue culture plates were infected at an MOI = 5 and allowed to grow for 12 h before replacing media with either tryptophan-replete or -deplete media. At 24 hpi, infected cells were washed in PBS and then cross-linked by the addition of 1% methanol-stabilized formaldehyde (Sigma Aldrich) for 30 mins at RT ˚C. Cross-linking was quenched by the addition of 250 mM glycine, pH 7.4 for 10 mins at RT ˚C. Samples were then washed with PBS to remove excess formaldehyde prior to being scraped into a total volume of 900 µL ice-cold Pierce RIPA buffer (ThermoFisher Scientific) + cOmplete EDTA-free protease inhibitor cocktail (Roche). The lysate was split into three 300 µL volumes (experimental IP, control IP, and input) and DNA was sheared by soni-cation in a water-cooled Bioruptor bath sonicator for a total of 90 cycles performed in two 45 min intervals of 30 s on/off at high power (Diagenode, Denville, New Jersey, USA). Sonicated lysates were cleared of cell debris by centrifugation at $21,100 \times g$, 4 ˚C for 5 min. Lysates were then cleared by incubation with 25 µL Pierce protein A/G magnetic beads (ThermoFisher Scientific) at 4 ˚C for 1.5 h with rotation. Input lysates were stored at −80 ˚C until nucleic acid purification.

Pierce protein A/G beads were loaded with antibody by first washing a 25 µL volume of beads in 350 µL TBS-T prior to being resuspended in 300 µL TBS-T with the addition of 20 µg of specific antibody (polyclonal rabbit α-TrpR, raised against the last 50 amino acids of *C. trachomatis* L2 TrpR, Li International) or control antibody (Normal Rabbit IgG, #2729, Cell Signaling Technology). Antibody-loaded beads were incubated for 1.5 h at 4 ˚C with rotation prior being magnetically separated and resuspended in their original volume of TBS-T. The antibody-loaded beads were then added to their respective lysates and incubated overnight at 4 ˚C with rotation.

Following overnight immunoprecipitation, beads were washed with 300 µL each of Low Salt, High Salt, and LiCl Immune Complex Wash Buffer (Sigma Aldrich) in sequence for 5 min at RT ˚C with rotation prior to elution of protein–DNA complexes in 400 µL of ice-cold Pierce IgG Elution Buffer, pH 2.0 (ThermoFisher Scientific). Eluates were immediately added to 80 µL of 1 M Tris, pH 8.5 to neutralize the solution. De-crosslinking was carried out by incubation of eluates at 65 ˚C for 6 h. Immunoprecipitated DNA was then purified from de-crosslinked eluates using the Macherey-Nagel NucleoSpin Gel and PCR Clean-up Kit (TakaraBio) following manufacturer protocols with the exception that input lysates were bound to the column in Buffer NTB due to the presence of SDS in the RIPA buffer. DNA was eluted in 40 µL Buffer NE and promoter enrichment was analyzed by RT-qPCR as described above. See Supplementary Data 2 for complete list of primers used for ChIP assays.

Promoter fold-enrichment was calculated as follows:

The $C_t$ values obtained from triplicate qPCR assays were averaged and dilution correction was calculated as:

$$C_{t(DF)} = C_t - \mathrm{Log}_2(DF)$$

where dilution factor equals the fold-dilution reported above (e.g., 50).

The percent recovery from ChIP-Input was calculated as follows:

$$\% \text{ Recovery} = 100 \times 2^{C_{t(Input)} - C_{t(ChIP)}}$$

where $C_{t(Input)}$ represents the $C_t$ value of the dilution-corrected ChIP-Input sample and $C_{t(ChIP)}$ represents the dilution-corrected $C_t$ value of the experimental or control antibody IPs.

The fold enrichment relative to the control antibody IP was calculated as follows:

$$\text{Fold enrichment} = \frac{\%\text{Recovery Experimental ChIP}}{\%\text{Recovery Control ChIP}}$$

These calculations assume 100% qPCR efficiency.

**Cloning**. Ligation independent cloning of pBOMBL-YtgCR vectors was performed by amplifying the YtgCR ORF from the *C. trachomatis* L2 434/Bu chromosome (WWW) or from another vector harboring the YYF variant, with a C-terminally appended FLAG sequence and 5′ KpnI and 3′ EagI restriction enzyme sites (See Supplementary Data 2 for complete list of primers used in this study). The PCR product was inserted into a pBOMB derivative, pBOMBL, using the HiFi DNA Assembly Kit (New England Biolabs, Ipswich, MA, USA). pBOMBL contains a modified ribosome binding site to reduce leaky expression. The complete characterization of this plasmid will be described in a forthcoming manuscript.

Cloning of the pET151-YtgCR-3xFLAG vectors was performed following manufacturer protocols for TOPO directional cloning into the pET151-D/TOPO base vector (Invitrogen, Thermo Fisher Scientific). The YtgCR sequence was amplified from *C. trachomatis* L2 gDNA with the reverse primer appending the 3xFLAG epitope through sequential amplifications since the entire sequence was not easily appended from one primer. The WWW motif was then mutated to YYF by base pair substitutions in the 5′-TGGTGGTGG-3′ sequence, converting it to 5′-TACTACTTC-3′. These sequences were cloned into another vector and then subcloned into the pET151 vector by adding the requisite CACC sequence to the 5′ end of the YtgCR ORF. All plasmids were sequence verified for proper insertion. Plasmids used in this study are described in Supplementary Table 1.

**Sequence analysis of WWW motifs**. All sequences from indicated chlamydial species were procured from the NCBI genome database, primarily by BLAST searches against the *C. trachomatis* L2 434/Bu CTL0174 or CTL0325 amino acid sequence. The fasta file containing all sequences used can be found in Supplementary Data 5. A multiple sequence alignment (MSA) of the YtgCR sequences was generated using the MUSCLE algorithm[70] in the MEGA7 software[71]. The MSA was graphically visualized in Jalview[72]. The full sequence alignments for CTL0174 and CTL0325 can be found in Supplementary Data 6 and 7, respectively.

**In silico analysis of putative TrpL sequences**. We analyzed the genomes of 16 bacterial species (acquired from the NCBI genome database) predicted to contain TrpL attenuators[62] for the purpose of comparison with the annotated TrpL in *C. trachomatis* (Supplementary Data 1). To identify attenuators, the intergenic region upstream of the *trp* operon where TrpL was predicted was manually surveyed for possible small upstream ORFs (uORF) containing tryptophan codons. Candidate uORFs had to meet the following criteria: 1.) contain an ATG start codon, 2.) contain tryptophan codons and 3.) contain an A/G-rich (≥50%) putative ribosome binding site (RBS) 4–14 bases upstream of the start codon. For some species, no candidate TrpL peptide or RBS could be identified. In the case of *Deinococcus radiodurans*, a candidate TrpL peptide was not identified where it had previously been predicted, though we note the possibility of a candidate TrpL found at a distal *trp* operon.

**Analysis of *C. trachomatis* L2 TrpL using PASIFIC**. The intergenic region upstream of the *trpBA* ORFs in *C. trachomatis*, along with the intergenic region upstream of the *E. coli* K12 (NCBI Accession: NC_000913.3) tr*pEDCBA* and *tnaAB* operons were input to the PASIFIC server (http://www.weizmann.ac.il/molgen/Sorek/PASIFIC/) for analysis using the default settings with the "I do not know the 3′ ends of the transcripts" option selected. The summary of results can be found in Supplementary Data 8, with the sequences used for analysis included in Supplementary Data 9.

**Graphs and statistical analysis**. All plots were made in the ggplot2 base package (version 3.1.0)[73] and the ggpubr package (version 0.2.3.) (https://CRAN.R-project.org/package=ggpubr) in R Studio (version 1.2.1335). Statistics were computed in R and specific tests are described in the figure legends. Figures were assembled in Adobe Illustrator (version 24.1.2) and Adobe Photoshop (version 24.1.2). For the model presented in Fig. 6, all elements were generated in Adobe Illustrator by the authors. Some elements have been used in prior publications by the authors[24,38] and were adapted under the Creative Commons license CC BY 4.0: https://creativecommons.org/licenses/by/4.0/.

**Reporting summary**. Further information on research design is available in the Nature Research Reporting Summary linked to this article.

## Data availability
Any data that support the findings of this study beyond what is included in the Supplementary Information are available from the corresponding author upon request. Requests for unique biological materials such as plasmids or transformants should be directed to the corresponding author. All genome sequence information was obtained from the National Center for Biotechnology Information (NCBI) Genome database (https://www.ncbi.nlm.nih.gov) and accession numbers to relevant genomes are referenced in the Methods where appropriate. Source data are provided with this paper.

## Code availability
All source code for each figure can be found in Supplementary Software or at https://github.com/npokorzynski/Pokorzynski-2020-Nat-Comm. Source code for the statistical analyses is included with the source code for each figure.

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

## Acknowledgements

The authors would like to acknowledge the members of the Carabeo lab for their support and critical feedback throughout the design and execution of this study. We thank the

Advanced Microscopy Core Facility staff at the University of Nebraska Medical Center for their excellent assistance in training and instrument maintenance. We would also like to acknowledge Dr. Rotem Sorek and Adi Millman (Weizmann Institute of Science, Israel) for their generous assistance in use of the PASIFIC tool. This study was supported by funding from the U.S. National Institutes of Health, National Institutes of Allergy and Infectious Disease grants R01 AI065545 (R.A.C.), R01 AI132406 (R.A.C. & S.P.O.) and F31 AI136295 (N.D.P.). This work was also supported by funding from the U.S. National Science Foundation through grant 1810599 (S.P.O.). N.D.P. was further supported by the National Institutes of Health Protein Biotechnology Training Program at Washington State University (T32 GM008336) and a fellowship from the Seattle Chapter of Achievement Rewards for College Scientists (ARCS). The University of Nebraska Medical Center Advanced Microscopy Core Facility receives partial support from the National Institute for General Medical Science (NIGMS) INBRE - P20 GM103427 and COBRE - P30 GM106397 grants, as well as support from the National Cancer Institute (NCI) for The Fred & Pamela Buffett Cancer Center Support Grant- P30 CA036727, and the Nebraska Research Initiative. This publication's contents and interpretations are the sole responsibility of the authors.

## Author contributions

N.D.P. and R.A.C. designed the experiments and wrote the manuscript. N.D.P. performed the experiments, and N.D.P. and R.A.C. analyzed the data. N.D.H. and S.P.O. provided reagents and samples critical to the completion of the research, including samples used for generating the data presented in Fig. 4b and the transformed chlamydial strains used in Fig. 2c–f. R.A.C. and S.P.O. both performed supervisory roles in the conception, direction, and completion of the research.

## Competing interests

The authors declare no competing interests.
