## [Peer Review File · Nature Communications]

REVIEWER COMMENTS

Reviewer #1 (Remarks to the Author):

This is an interesting manuscript that describes a new mechanism to regulate tryptophan biosynthesis/salvage in *Chlamydia trachomatis*. Regulation of expression of tryptophan biosynthetic genes in bacteria has been studied extensively for many years, including several variations on transcription attenuation. However, it has been quite a while since a fundamentally different variation has been described. In this work transcription of *trpBA* is shown to be controlled by both a tryptophan-dependent DNA-binding repressor protein TrpR at the major promoter for *trpBA*, as well as by an iron-dependent repressor YtgR that controls both transcription initiation from an alternative promoter and read-through from the upstream major promoter. This system of regulation tying tryptophan and iron-mediated responses is novel and would be of interest to others in the fields of transcription regulation as well as bacterial pathogenesis.

The manuscript is generally well written but given the complexity of this control mechanism that responds to both tryptophan and iron it can be hard to follow. I would have appreciated a bit more introduction to *Chlamydia trachomatis* and its role in health. I would suggest separating the schematic diagram of the genes and promoters involved (currently fig 1 d) into the first figure of the paper, or at least move it to Fig. 1a, including indications of the roles of TrpR and YtgR rather than embedding it as part of Figure 1d, especially given that it is the first figure called to in the body of the text.

On line 129-30 as well as later statements on lines 386-87 indicate that transcription factor Rho plays an important role in transcription attenuation control of the *E. coli* *trp* operon. This is incorrect, in *E. coli* attenuation is mediated by an intrinsic terminator RNA structure (termed 3:4). Thus the parallels the authors draw between the roles of Rho in both systems are not valid.

Line 190-92 Explain why you expect tryptophan starvation to have the maximal effect i.e. the role of TrpR, and explain why iron limitation would be less so.

Figure 1C, why use values on the Y-axis such that you only use less than 1/3 of the scale to plot your data?

Line 211-12 Fig.1 d states that "transient iron limitation specifically induced expression of *trpBA* while *trpR* expression is unchanged." The differences in the effects of the 8+6 Bpd treatment on expression these genes does not appear to warrant such a definitive statement. *trpR* expression does increase slightly and *trpBA* expression is only slightly greater than *trpR*.

Why are three different means used to assess YtgR expression in this work? In Fig. 2b immunoblotting for YtrG, in Fig. 2e immunofluorescence of antibodies against FLAG, and in Fig 3 immunoblotting of the FLAG tag on YtrG.

What is gained from the confocal microscopy in Fig. 2c? Localization isn't mentioned?

Line 292 the phrase "up to six tryptophan codons" doesn't make sense. Just state how many are present in the gene.

Line 318 the statements about YtgCR cleavage in *E. coli* in are confusing. It's not clear why this is relevant nor why it is different in minimal medium, and it isn't clear as compared to what other medium

Fig. 3 It would make more sense to present the indolmycin panels (currently c and d) first and then the control ANM3365 (currently a and b).

Line 414. It isn't clear what ytgCR polarization means as compared to ytg operon polarization (Fig. 4c)

Line 486-8 this sentence doesn't make sense, particularly the statement "this attenuation functions in trans" that isn't possible. It may be controlled by a trans-acting factor but it functions in cis to control the downstream genes.

Fig. 6. What does the orange star represent?

Line 540 spelling issue

Reviewer #2 (Remarks to the Author):

Pokorzynski, Hatch, Ouellette and Carabeo report that the iron-dependent repressor YtgR of *Chlamydia trachomatis* also functions as a tryptophan-dependent regulator. This study is of high impact and will be of interest to the microbiology field because it describes a new mechanism of trp operon attenuation that differs from both the classical Yanofsky mechanism and the *Bacillus subtilis* mechanism that involves a trans-factor called TRAP. In the Pokorzynski mechanism, the trans-factor is a transcription factor YtgR that is tryptophan-dependent because it contains an unusual WWW coding motif that limits its production during low tryptophan conditions. Because this transcription factor YtgR is an iron-dependent repressor, there appears to be cross-talk between tryptophan- and iron-dependent gene regulation in *C. trachomatis*, which will make the study of interest in the *Chlamydia* field. The study builds upon previous excellent work from this group to describe the mechanism by which *C. trachomatis* can respond to a tryptophan- and iron-poor environment to salvage indole from the microbiota of the female lower genital tract to synthesize tryptophan. The study was generally well performed but could be improved by some quantifications and controls as discussed below, and a more complete discussion.

In this study, the authors provided data for 3 main elements of their novel model:

1. YtgR-dependent repression of trpBA is dependent on tryptophan availability.
 - a. RT-PCR data showed that trpBA repression was dependent on tryptophan availability – the effect was hypothesized, but not shown, to be due to YtgR, which is known to regulate the trpBA promoter. Another theoretical possibility is that the effect is through TrpR, which is tryptophan-dependent. Has it been confirmed that TrpR does not regulate the trpBA promoter?
 - b. 5'-RACE analysis of trpR vs trpBA transcription with tryptophan and iron depletion: the results should be shown as trpB and trpR transcript levels, and not just as a trpB/trpR ratio because some interventions may affect both trpB and trpR transcript levels. These quantifications would also highlight that that the increase in trpBA transcription is greater with iron depletion compared to tryptophan depletion, which should be discussed. Since trpR is barely detectable with 24 hour BDP in Fig 5b, why is the trpB/trpR ratio not higher in Fig 5a?

2. YtgR protein levels are dependent on tryptophan availability.

Western blot levels of YtgR were lower with tryptophan limitation but the effect was relatively modest and somewhat overstated in the text – these levels should be quantified. The indole rescue experiment was a good way to show that decreased protein levels was dependent on tryptophan availability.

3. YtgR protein levels are dependent on the WWW motif

The YtgCR WWW vs YYF experiment is good but how sensitive and reliable is the MFI assay for measuring protein levels? The assay should have a negative control with antibody to another protein to show that it is not affected by the WWW motif in YtgR. The effect of the WWW motif on YtgR levels and control proteins could be shown and quantified by Western blots. The indolmycin

experiment was good for showing that translation of YtgR is dependent on tryptophanyl-tRNA synthetase, but how do the authors know that the result was due to substitution of the WWW motif and not because the number of tryptophan residues in YtgR was reduced by half?

Additional comments

1. Where does the 18hpi trpBA transcript come from in Fig 5b? There is no trpR transcript so it should be transcribed from the trpBA promoter. But YtgR is present at 12 and 24 hpi by Western blot in Fig 2b, so shouldn't it repress the trpBA promoter? And what happens between 18 and 24 hpi when the trpR transcript appears? The manuscript would be strengthened by discussing these regulatory changes in transcription from the two trp promoters during the chlamydial developmental cycle.
2. Figure 3a and 3c. What is the 100 kD band that also changes with AN3365 and indolmycin. Is it full length YtgCR?
3. Figure 3. In light of the growth defect with transformed strains, how were cell lysates normalized so that they could be compared?
4. Show primer location in Fig 4a.
5. The YYF mutant could be used to study the author's hypothesis that the WWW motif within YtfCR is responsible for the polarized expression of the ytgABCD operon.
6. *C. ibidis* only has 1 of the 3 W residues in YtgR. How does this affect the attenuation model?
7. The tryptophan depletion part of Fig 6 is hard to follow. Label ribosome. This figure doesn't do justice to the model because it is mostly focused on tryptophan regulation. It doesn't show the larger context, which is that tryptophan controls iron regulation in Chlamydia to go along with the author's previous discovery that iron controls tryptophan regulation. The discussion should address the significance of this cross-talk for the Chlamydia infection.
8. Suggest a figure to illustrate how this novel attenuation mechanism compares and contrasts to attenuation mechanisms in *E. coli* and *Bacillus*.

REVIEWER COMMENTS

Responses in BOLD

Reviewer #1 (Remarks to the Author):

This is an interesting manuscript that describes a new mechanism to regulate tryptophan biosynthesis/salvage in *Chlamydia trachomatis*. Regulation of expression of tryptophan biosynthetic genes in bacteria has been studied extensively for many years, including several variations on transcription attenuation. However, it has been quite a while since a fundamentally different variation has been described. In this work transcription of *trpBA* is shown to be controlled by both a tryptophan-dependent DNA-binding repressor protein TrpR at the major promoter for *trpBA*, as well as by an iron-dependent repressor YtgR that controls both transcription initiation from an alternative promoter and read-through from the upstream major promoter. This system of regulation tying tryptophan and iron-mediated responses is novel and would be of interest to others in the fields of transcription regulation as well as bacterial pathogenesis.

The manuscript is generally well written but given the complexity of this control mechanism that responds to both tryptophan and iron it can be hard to follow. I would have appreciated a bit more introduction to *Chlamydia trachomatis* and its role in health. I would suggest separating the schematic diagram of the genes and promoters involved (currently fig 1 d) into the first figure of the paper, or at least move it to Fig. 1a, including indications of the roles of TrpR and YtgR rather than embedding it as part of Figure 1d, especially given that it is the first figure called to in the body of the text.

To address the concerns of the reviewer regarding Figure 1d, we have removed reference to it in the introduction (i.e. it is no longer the first figure called to in the text; Line 140) and indicated that P_{trpR} is TrpR-regulated and that P_{trpBA} is YtgR-regulated in the figure. We kept the figure in the same place to maintain the logical flow with the schematic immediately preceding the relevant data, i.e. the *trpRBA* RT-qPCR data.

On line 129-30 as well as later statements on lines 386-87 indicate that transcription factor Rho plays an important role in transcription attenuation control of the *E. coli* *trp* operon. This is incorrect, in *E. coli* attenuation is mediated by an intrinsic terminator RNA structure (termed 3:4). Thus the parallels the authors draw between the roles of Rho in both systems are not valid.

In the relevant passage noted above, we cited two publications related to the regulation of transcription termination by Rho in the leader sequence of the *E. coli* *trp* operon:

In the first publication (Korn & Yanofsky, *J Mol Biol*, 1976), the authors describe that a previously characterized class of *E. coli* polarity suppressor mutants (i.e. mutants that display increased expression of the structural genes of the *trpEDCBA* operon) possess defective Rho factor that has reduced transcription termination activity and altered biochemical properties. The major conclusion from these findings is that Rho contributes to transcription termination in the leader sequence *in vivo*.

In the second publication (Fuller & Platt, *Nuc Acids Res*, 1978), the authors show that Rho factor is not required for efficient transcription termination *in vitro*, but that Rho increases synthesis of the leader mRNA in *in vitro* transcription assays. They conclude that this is likely due to Rho dissociating RNA polymerase from the template to allow

new polymerase molecules to begin transcription. They conclude from these findings that the principal function of Rho in the attenuator is to dissociate RNA polymerase from the template following the formation of a termination complex and does not itself promote transcription termination.

Indeed, in one of the seminal publications describing the mechanism of translational control of attenuation in the leader sequence from Charles Yanofsky's group (Zurawski *et al.*, *PNAS*, 1978), the model presented in Figure 3 specifically includes a role for Rho in dissociating RNA polymerase and references Fuller & Platt, 1978.

After reviewing the literature further, we have observed that despite these data, the Yanofsky group began to favor a Rho-independent model of transcription termination as more evidence emerged that the terminator/anti-terminator RNA secondary structure along with ribosomal stalling could efficiently promote transcription termination *in vitro*. For example, already in 1981, Yanofsky cast doubt on the role of Rho in a review article in *Nature*: "Finally, mutations affecting RNA polymerase and the transcription termination protein rho also affect termination at the *trp* attenuator. The role of rho, if it has one, in transcription termination at the *trp* attenuator is unclear." This seems to contradict the direct genetic evidence provided in 1976 for the involvement of rho in transcription termination of the *trp* operon. By 1985, with evidence that translation of the leader peptide directly influences RNA polymerase pausing, Landick *et al.* (*PNAS*) suggested that this coupling of translation and transcription may prevent the synthesis of an mRNA species suitable for Rho-mediated transcription termination. However, no direct evidence was presented to support this hypothesis. As far as we can tell from reviewing the literature, this is the last reference to the role of Rho in transcription termination of the *trp* leader sequence in *E. coli*.

To our knowledge, no evidence definitively excluded the possibility that rho is involved in transcription termination of the *E. coli trp* leader sequence. Thus, as it stands, the evidence supports a role for rho in dissociating RNA polymerase from the pre-formed termination complex. We have therefore chosen to edit lines 129-30 and 388-89 to more accurately reflect the data regarding the role of rho in termination at the attenuator in *E. coli*. If the Reviewer is aware of a publication that addresses the role of rho in attenuation and demonstrates that it is not involved, we would be happy to modify the text as necessary.

Line 190-92 Explain why you expect tryptophan starvation to have the maximal effect i.e. the role of TrpR, and explain why iron limitation would be less so.

In lines 190-93, we have edited the sentences to more directly outline our hypothesis and expectations. We note that the original sentence indicated that tryptophan starvation would result in maximal expression of the *trpRBA* operon only if it simultaneously inactivated TrpR and YtgR repression. Iron depletion would be expected to result in lower levels of expression since it only inactivates YtgR repression. The sentences now read, "If tryptophan limitation inactivated both TrpR and YtgR repression of *trpBA*, we expect maximal expression of *trpBA* during tryptophan starvation. A corollary is that the effect of iron limitation, which only inactivates YtgR repression, would result in elevated but not maximal transcription of *trpBA*."

Figure 1C, why use values on the Y-axis such that you only use less than 1/3 of the scale to plot

your data?

The scales for the RT-qPCR plots were all set to the same scale used for the *trpRBA* data, which is much more strongly induced by these starvation conditions. We felt that it was useful to keep the scales comparable so that the reader could directly compare the magnitude of expression change between the genes assayed.

Line 211-12 Fig.1 d states that “transient iron limitation specifically induced expression of *trpBA* while *trpR* expression is unchanged.” The differences in the effects of the 8+6 Bpd treatment on expression these genes does not appear to warrant such a definitive statement. *trpR* expression does increase slightly and *trpBA* expression is only slightly greater than *trpR*.

In lines 212-13 we have tempered our conclusions from these data and described them in a way that reflects the non-significant difference in the change of *trpR* expression. The sentence now reads, “As expected, transient iron starvation specifically induced the expression of *trpBA*, while *trpR* expression was statistically indistinguishable from mock treatment.”

Why are three different means used to assess YtgR expression in this work? In Fig. 2b immunoblotting for YtrG, in Fig. 2e immunofluorescence of antibodies against FLAG, and in Fig 3 immunoblotting of the FLAG tag on YtrG.

The three different experiments that assayed YtgR expression were performed to assess different aspects of the model:

- 1. The immunoblot in Figure 2b tested the general hypothesis that YtgR was regulated by tryptophan availability.**
- 2. The IFA assay tested the more specific hypothesis that the WWW motif is involved in regulating YtgR tryptophan-sensitivity.**
- 3. The *E. coli* expression system tested the hypothesis that the charging of tryptophanyl-tRNA and subsequently inhibition of translation at the WWW motif was the mechanism by which YtgR levels are sensitive to tryptophan limitation.**

What is gained from the confocal microscopy in Fig. 2c? Localization isn't mentioned?

The microscopy in Figure 2c-d is meant to provide representative images from which the data in Figure 2e-f is derived. With respect to localization, this experiment was not designed to investigate this aspect of YtgCR, but to quantify fluorescence using a technique that minimizes background.

Line 292 the phrase “up to six tryptophan codons” doesn't make sense. Just state how many are present in the gene.

Lines 291-92 were modified slightly to reflect that we cannot know for certain the exact number of tryptophan codons in the YtgR domain given that we currently do not know the location of cleavage site that liberates YtgR from YtgCR. The phrase “up to six” was meant to imply that based on the expected size of the cleavage product, reported here and in our previous publications, we expect that there may be as many as 6 tryptophan codons in the cleavage product. The sentence now reads, “This is notable given that the YtgR domain itself is tryptophan-rich and may encode six tryptophan codons based on the observed molecular weight of the cleavage product.”

Line 318 the statements about YtgCR cleavage in *E. coli* in are confusing. It's not clear why this is relevant nor why it is different in minimal medium, and it isn't clear as compared to what other medium

This statement was included to provide context to the reader given that in our previous work (Thompson *et al.* 2012. *PNAS*) we monitored YtgCR cleavage in *E. coli* cultured in rich media (LB) and observed multiple cleavage bands. Therefore, we felt that it was useful to indicate to the reader why the cleavage pattern is different in this experiment. We have edited line 317 to indicate that the difference is likely due to culturing in different media conditions. The sentences now read, "As in *Ctr*, YtgCR expressed in *E. coli* grown in rich media is cleaved and typically yields multiple cleavage products, suggesting non-specific cleavage due to dysregulated or off-target proteolytic activity. However, in minimal media, we observed that cleavage of YtgCR yielded one major band at the expected molecular weight of YtgR (30 kDa)."

Fig. 3 It would make more sense to present the indolmycin panels (currently c and d) first and then the control ANM3365 (currently a and b).

The AN3365 data is presented first in Figure 3 for two reasons:

- 1. We felt that it flowed better in the text to describe that general translation inhibition by AN3365 treatment reduces YtgR expression irrespective of the WWW motif.**
- 2. The AN3365 treatment is both a reference condition and a control, and therefore the reader can interpret the indolmycin data (specific inhibition at tryptophan codons) in reference to the AN3365 data (general inhibition at the much more abundant leucine codons).**

Line 414. It isn't clear what ytgCR polarization means as compared to ytg operon polarization (Fig. 4c)

To address this concern, we have included a paragraph in the discussion that covers the implications of these findings and highlights how the two independent termination mechanisms (that of the ytgCR ORF and that of the entire operon, e.g. ytgD:ytgA) may function. Please refer to Lines 516-28.

Line 486-8 this sentence doesn't make sense, particularly the statement "this attenuation functions in trans" that isn't possible. It may be controlled by a trans-acting factor but it functions in cis to control the downstream genes.

We have edited the text (Lines 489-90) to indicate that the attenuation does not "function" in trans but is mediated by a trans-acting factor. The sentence now reads, "This regulatory mechanism parallels *cis*-attenuation by TrpL in *E. coli*, though notable differences exist: namely, this attenuation mechanism is mediated by a *trans*-acting transcription factor."

Fig. 6. What does the orange star represent?

The orange star is meant to imply transcription termination, but we have removed it for clarity.

Line 540 spelling issue

This spelling issue has been corrected.

Reviewer #2 (Remarks to the Author):

Pokorzynski, Hatch, Ouellette and Carabeo report that the iron-dependent repressor YtgR of *Chlamydia trachomatis* also functions as a tryptophan-dependent regulator. This study is of high impact and will be of interest to the microbiology field because it describes a new mechanism of trp operon attenuation that differs from both the classical Yanofsky mechanism and the *Bacillus subtilis* mechanism that involves a trans-factor called TRAP. In the Pokorzynski mechanism, the trans-factor is a transcription factor YtgR that is tryptophan-dependent because it contains an unusual WWW coding motif that limits its production during low tryptophan conditions. Because this transcription factor YtgR is an iron-dependent repressor, there appears to be cross-talk between tryptophan- and iron-dependent gene regulation in *C. trachomatis*, which will make the study of interest in the *Chlamydia* field. The study builds upon previous excellent work from this group to describe the mechanism by which *C. trachomatis* can respond to a tryptophan- and iron-poor environment to salvage indole from the microbiota of the female lower genital tract to synthesize tryptophan. The study was generally well performed but could be improved by some quantifications and controls as discussed below, and a more complete discussion.

In this study, the authors provided data for 3 main elements of their novel model:

1. YtgR-dependent repression of trpBA is dependent on tryptophan availability.
 - a. RT-PCR data showed that trpBA repression was dependent on tryptophan availability – the effect was hypothesized, but not shown, to be due to YtgR, which is known to regulate the trpBA promoter. Another theoretical possibility is that the effect is through TrpR, which is tryptophan-dependent. Has it been confirmed that TrpR does not regulate the trpBA promoter?

Akers & Tan (*J Bacteriol*, 2006) reported an unpublished observation that in their hands TrpR was unable to bind to any region of the *trpBA* promoter. We cited this observation in the introduction (see Line 142). We have confirmed these findings with a targeted chromatin immunoprecipitation-qPCR (ChIP-qPCR) assay using a polyclonal antibody generated against chlamydial TrpR (Lines 448-51; Supplementary Fig 8). These data demonstrate that TrpR binds to the *trpR* promoter in a tryptophan-dependent manner, but not to the *trpBA* promoter.

- b. 5'-RACE analysis of trpR vs trpBA transcription with tryptophan and iron depletion: the results should be shown as trpB and trpR transcript levels, and not just as a trpB/trpR ratio because some interventions may affect both trpB and trpR transcript levels. These quantifications would also highlight that the increase in trpBA transcription is greater with iron depletion compared to tryptophan depletion, which should be discussed. Since trpR is barely detectable with 24 hour BDP in Fig 5b, why is the trpB/trpR ratio not higher in Fig 5a?

In Figure 5a, we calculated a *trpB:trpR* ratio as a proxy measurement of *P_{trpBA}* activity, which we distinguish from the level of expression of either gene. This assay reports on *trpBA* transcription events that occur independently of the *trpRBA* transcription event initiated at *P_{trpR}* (indicated by a ratio >1.0). These data were calculated from the Ct values generated for the data presented in Figure 1d, and therefore the expression of the individual genes under these conditions have been addressed with those data.

Regarding the relative increase in *trpBA* expression under iron vs. tryptophan depleted conditions, our model suggests that *trpBA* should be maximally expressed during tryptophan depletion, where both TrpR and YtgR are inactivated, which is indeed what we observe.

With respect to the presence of the TSS at *P_{trpR}* in Figure 5b, we emphasize that the RACE data should not be interpreted as an indication of expression levels, as these data are generated from PCR-amplified cDNAs and are therefore not quantitative. Additionally, it is important to note that absence of a product in the RACE data does not imply absolute absence. This highlights the importance of the quantitative data included in Figure 5a. Ultimately, however, the data presented in Figure 5b were only intended to introduce the more precise and quantitative determination of the *P_{trpBA}* TSS location in Figures 5c-d.

2. YtgR protein levels are dependent on tryptophan availability.
Western blot levels of YtgR were lower with tryptophan limitation but the effect was relatively modest and somewhat overstated in the text – these levels should be quantified. The indole rescue experiment was a good way to show that decreased protein levels was dependent on tryptophan availability.

We have quantified the levels of YtgR and OmpA expression normalized to GroEL_1, which lacks any tryptophan codons and is therefore the most appropriate control band for normalization. These quantifications have been provided in Supplementary Figure 3 (Line 272).

3. YtgR protein levels are dependent on the WWW motif
The YtgCR WWW vs YYF experiment is good but how sensitive and reliable is the MFI assay for measuring protein levels?

We refer the reviewer to Supplementary Figure 4a-b, which shows that in the absence of aTc there is no discernable FLAG signal within the chlamydial inclusion of the transformants. To support these data, we have quantified FLAG signal from uninduced samples that were prepared as technical controls for the data presented in Figure 2c-f (Lines 294-95; Supplementary Figure 4c-d). These data show that in the absence of aTc, FLAG signal is negligible and that the FLAG signal we detect is specific to the induced expression of YtgCR-FLAG.

The assay should have a negative control with antibody to another protein to show that it is not affected by the WWW motif in YtgR. The effect of the WWW motif on YtgR levels and control proteins could be shown and quantified by Western blots.

To address this concern, we have quantified the GroEL signal used to generate a mask for inclusions in this assay (Lines 292-293; Figure 2f). These data demonstrate the GroEL expression is relatively unaffected by tryptophan availability. Because we have used immunoblotting to investigate the tryptophan sensitivity of YtgR levels in Figure 2b, we believe that the IFA assay is more appropriate as it incorporates inter-inclusion heterogeneity which is important not only for monitoring changes in response to nutrient limitation – the effect of which is heterogeneous itself – but also because we are relying on chlamydial strains transformed with plasmids in the presence of penicillin, which introduces more heterogeneity that may interfere with population-level assays such as immunoblotting.

The indolmycin experiment was good for showing that translation of YtgR is dependent on tryptophanyl-tRNA synthetase, but how do the authors know that the result was due to substitution of the WWW motif and not because the number of tryptophan residues in YtgR was reduced by half?

We believe this to be a misinterpretation (see above concerning Line 292). The amount of tryptophan codons in YtgCR was only reduced by ~27% (3/11). The tryptophan codons encoded within the YtgR domain were unaltered. We believe this underscores the important regulatory role of the WWW motif given that the YtgR domain is itself highly enriched for tryptophan codons but is still relatively well expressed in the presence of indolmycin if the WWW motif is mutated (Figure 3c-d).

Additional comments

1. Where does the 18hpi trpBA transcript come from in Fig 5b? There is no trpR transcript so it should be transcribed from the trpBA promoter. But YtgR is present at 12 and 24 hpi by Western blot in Fig 2b, so shouldn't it repress the trpBA promoter? And what happens between 18 and 24 hpi when the trpR transcript appears? The manuscript would be strengthened by discussing these regulatory changes in transcription from the two trp promoters during the chlamydial developmental cycle.

Please see the response above regarding the interpretation of the 5'-RACE data (Point 1b).

2. Figure 3a and 3c. What is the 100 kD band that also changes with AN3365 and indolmycin. Is it full length YtgCR?

We suspect that the 100 kDa species may be YtgCR dimers, however we are hesitant to indicate this given that these are denaturing PAGE gels and that we have not directly addressed this possibility.

3. Figure 3. In light of the growth defect with transformed strains, how were cell lysates normalized so that they could be compared?

As noted in the methods ("Expression of YtgCR in *E. coli*"), the volume of lysate loaded into each well was normalized by the final OD of the respective sample such that each well received the same OD amount of lysate.

4. Show primer location in Fig 4a.

The primer location has been added to Figure 4a.

5. The YYF mutant could be used to study the author's hypothesis that the WWW motif within YtgCR is responsible for the polarized expression of the ytgABCD operon.

Such population-level assays (e.g. RT-qPCR) would likely require stable chromosomal alterations to *ytgCR* to be reliable and interpretable. As it stands, our transformed strains do not afford us the opportunity to be able to distinguish between the chromosomal copy of *ytgCR* and the plasmid-encoded copy by RT-qPCR. At present, genetic approaches for introducing chromosomal mutations in *Chlamydia* are not well optimized for genes within operons as they require the introduction of resistance markers and therefore may

have unpredictable polar effects. It is for these reasons and others that we included several different WWW motif controls in these experiments to reinforce the role of the WWW motif in mediating these phenotypes.

6. *C. ibidis* only has 1 of the 3 W residues in YtgR. How does this affect the attenuation model?

Many chlamydial species lack a *trp* operon, and this is apparently the case for *C. ibidis* as well, as they do not encode a TrpB ortholog identifiable by BLAST (personal observation). As noted in the text, we propose that the lack of complete WWW motifs in avian and so-called “environmental” chlamydiae suggests that the WWW motif evolved in adaptation to mammalian hosts where maintaining tight regulation of the *trp* operon was advantageous. This seems most important for *C. trachomatis*, which is the only species we are aware of that contains the *trpRBA* operon with the intergenic region that harbors the cis-regulatory YtgR operator sequence. It is possible, however, that the WWW motif of YtgCR functions to compensate for the lack of TrpR in other species such that they can maintain tryptophan-dependent regulation of other genes by YtgR repression.

7. The tryptophan depletion part of Fig 6 is hard to follow. Label ribosome. This figure doesn't do justice to the model because it is mostly focused on tryptophan regulation. It doesn't show the larger context, which is that tryptophan controls iron regulation in Chlamydia to go along with the author's previous discovery that iron controls tryptophan regulation. The discussion should address the significance of this cross-talk for the Chlamydia infection.

We have created a more comprehensive model figure that incorporates our working model for YtgR-mediated attenuation of the *trpRBA* operon within the larger biological context. In addition, we have added a paragraph to the discussion that places YtgCR regulation by the WWW motif within the broader biological context of chlamydial infection in the female lower genital tract (Lines 531-44).

8. Suggest a figure to illustrate how this novel attenuation mechanism compares and contrasts to attenuation mechanisms in *E. coli* and *Bacillus*.

We believe that the suggested figure may be better suited for a review article or similar, in that we have not provided data that directly compares the three attenuation mechanisms.

REVIEWER COMMENTS

Reviewer #1 (Remarks to the Author):

Below are my comments to some of the responses of the author to the initial review.

I still feel that if this is an interesting paper but if it is going to be accessible to a more general audience, such as would be expected from Nature Communications, that the introduction needs improvement. For example, I suggested more background information about *Chlamydia trachomatis*, the means of growing it and its role in health, which was not addressed in the authors response.

The response about the potential role of Rho in controlling the *E. coli* trp operon is concerning. The authors cite papers from the Yanofsky lab from 1976 and 1978 (both over 40 years old) and then follows up with a statement "that after these data, the Yanofsky group began to favor a Rho-independent model of transcription termination as more evidence emerged" Yet the author then concludes that "To our knowledge, no evidence definitively excluded the possibility that Rho is involved in transcription termination of the *E. coli* trp leader sequence." Conclusively proving negative is indeed difficult, especially in this case when the gene for Rho is essential in *E. coli*. Nevertheless, the preponderance of evidence strongly supports the view that the 3:4 RNA structure in the *E. coli* trp leader region is an intrinsic transcription terminator, in fact it is often described as a classic model for an intrinsic terminator being a "GC rich stem loop followed by an uninterrupted run of Us." The most compelling evidence for this is generally thought to be DNA heteroduplex studies by Ryan and Chamberlin (JBC 258, 4690-93 (1983)). The trp attenuator has been described and Rho independent in the definitive reviews by Landick and Yanofsky in 1987, and Landick and Turnbough 1992, who state "Evidence for the role of alternative RNA secondary structures in controlling attenuation is compelling" and genetics textbooks it as an intrinsic terminator. Perpetuation a long-disbelieved interpretation of the mechanism of attenuation in *E. coli* is a disservice to the community.

Why are three different means used to assess YtgR expression in this work?

The three different experiments that assayed YtgR expression were performed to assess different aspects of the model:

1. The immunoblot in Figure 2b tested the general hypothesis that YtgR was regulated by tryptophan availability.
2. The IFA assay tested the more specific hypothesis that the WWW motif is involved in regulating YtgR tryptophan-sensitivity.
3. The *E. coli* expression system tested the hypothesis that the charging of tryptophanyl-tRNA and subsequently inhibition of translation at the WWW motif was the mechanism by which YtgR levels are sensitive to tryptophan limitation.

While this explains the purpose of the three different experiments, it does not explain why the three different methods were used nor why any of the methods better answer the questions being asked.

What is gained from the confocal microscopy in Fig. 2c? Localization isn't mentioned?

The microscopy in Figure 2c-d is meant to provide representative images from which the data in Figure 2e-f is derived.

Again, this does not answer the question of what is gained from doing confocal microscopy to quantify expression of a protein.

Line 318 the statements about YtgCR cleavage in E. coli in are confusing. It's not clear why this is relevant nor why it is different in minimal medium, and it isn't clear as compared to what other medium.

The sentences now read, "As in Ctr, YtgCR expressed in E. coli grown in rich media is cleaved and typically yields multiple cleavage products, suggesting non-specific cleavage due to dysregulated or off-target proteolytic activity. However, in minimal media, we observed that cleavage of YtgCR yielded one major band at the expected molecular weight of YtgR (30 kDa)."

Why would proteolysis be different in rich media vs minimal? I still find this an odd topic to be focused on in this paper and at best it is distracting and doesn't really add to the discussion of the mechanism of attenuation.

Reviewer #2 (Remarks to the Author):

The revised manuscript and rebuttal have addressed a number of points from the initial review. I have just a few unresolved issues about the different assays to measure YtgR levels, which were a concern for both reviewers. The authors provided explanations for the selective use of Western blots vs IFA assay, but there remains suspicion that the IFA assay was used because the Western blots were not sensitive enough to show a difference.

1. The IFA assay results remain unconvincing. There was a wide range of signal strength, with almost complete overlap between Trp+ and Trp- conditions for both YtgR and the GroEL control, and modest difference in the population median. What was the fold change in the population mean, and was this statistically significant? Why were violin plots used in Fig 2e-f but boxplots in Supp Fig 4c-d?

2. The YtgR Western blot with tryptophan limitation should have better normalization. The text describes YtgR and OmpA downregulation compared to GroEL_1, as well as unhindered GroEL_1 expression. But the latter effect cannot be seen since YtgR levels were normalized to GroEL_1. Levels of these 3 proteins should be normalized to another protein.

RESPONSE TO REVIEWER COMMENTS

Where applicable, original reviewer comments are left in normal text, while original responses are in bold and the second reviewer comments are underlined. All responses for the second round of revision are italicized.

Reviewer #1 (Remarks to the Author):

Below are my comments to some of the responses of the author to the initial review.

Issue 1: I still feel that if this is an interesting paper but if it is going to be accessible to a more general audience, such as would be expected from Nature Communications, that the introduction needs improvement. For example, I suggested more background information about *Chlamydia trachomatis*, the means of growing it and its role in health, which was not addressed in the authors response.

*We apologize for this oversight. We have now included background on *Chlamydia*, its developmental cycle, and effects of nutrient deprivation on growth and transcriptional response, with an emphasis on tryptophan and iron. Please see lines 119-50 for details.*

Issue 2: The response about the potential role of Rho in controlling the *E. coli* trp operon is concerning. The authors cite papers from the Yanofsky lab from 1976 and 1978 (both over 40 years old) and then follows up with a statement “that after these data, the Yanofsky group began to favor a Rho-independent model of transcription termination as more evidence emerged” Yet the author then concludes that “To our knowledge, no evidence definitively excluded the possibility that Rho is involved in transcription termination of the *E. coli* trp leader sequence.” Conclusively proving negative is indeed difficult, especially in this case when the gene for Rho is essential in *E. coli*. Nevertheless, the preponderance of evidence strongly supports the view that the 3:4 RNA structure in the *E. coli* trp leader region is an intrinsic transcription terminator, in fact it is often described as a classic model for an intrinsic terminator being a “GC rich stem loop followed by an uninterrupted run of Us.” The most compelling evidence for this is generally thought to be DNA heteroduplex studies by Ryan and Chamberlin (JBC 258, 4690-93 (1983)). The trp attenuator has been described and Rho independent in the definitive reviews by Landick and Yanofsky in 1987, and Landick and Turnbough 1992, who state “Evidence for the role of alternative RNA secondary structures in controlling attenuation is compelling” and genetics textbooks it as an intrinsic terminator. Perpetuation a long-disbelieved interpretation of the mechanism of attenuation in *E. coli* is a disservice to the community.

We did not mean to offend the Reviewer by retaining the statement. We were under the impression that providing the appropriate references and clarifying our point would be sufficient. While researching the literature on this topic, we were perplexed by the drastic shift from a Rho-dependent model to an RNA structure-mediated termination, without any publications that definitively excluded or refuted the Rho-dependent model. This was the point we wanted to get across in our initial response. Regardless, we have now updated the manuscript and removed the offending references to Rho-dependent termination and the trp operon and modified the text accordingly to remove any suggestions of parallels in this regard.

Issue 3: Why are three different means used to assess YtgR expression in this work?

The three different experiments that assayed YtgR expression were performed to assess different aspects of the model:

1. The immunoblot in Figure 2b tested the general hypothesis that YtgR was regulated by tryptophan availability.
2. The IFA assay tested the more specific hypothesis that the WWW motif is involved in regulating YtgR tryptophan-sensitivity.
3. The *E. coli* expression system tested the hypothesis that the charging of tryptophanyl-tRNA and subsequently inhibition of translation at the WWW motif was the mechanism by which YtgR levels are sensitive to tryptophan limitation.

While this explains the purpose of the three different experiments, it does not explain why the three different methods were used nor why any of the methods better answer the questions being asked.

We apologize for this oversight; we misinterpreted the intention of the reviewers questions. To answer the questions as to why various techniques were used, the reason for the different means of monitoring YtgR expression level is the differences in biological and experimental contexts. To elaborate, in Figure 2b, we were testing if endogenous YtgR levels would respond to transient tryptophan depletions. We felt that it was necessary that this question was answered using the endogenous YtgR to establish that YtgR was regulated by tryptophan levels in a genetically unmodified experimental system.

In Figure 2c-e, we were attempting to determine if tryptophan-sensitivity of YtgR could be accounted for by the WWW motif alone. As noted in the manuscript (Lines 291-92), endogenous YtgR expression is quite low and difficult to detect until late time points in infection, which can be observed in the immunoblot in Fig. 2b. This is problematic given the biphasic developmental cycle of chlamydia, where late cycle inclusions are primarily occupied by infectious EBs, which exhibit reduced metabolic activity. Therefore, addressing this question in a robust manner required that YtgCR with the WWW-to-YYF substitution be expressed from an inducible plasmid; and for thoroughness, the YtgCR version as well. Ideally, the substitution could have been done to the chromosomal copy, which in Chlamydia genetics would have required allelic recombination. The current method, which is still inefficient, involves the retention of a marker, e.g. antibiotic resistance that would be maintained and incorporated into the genome to identify successful allelic recombinants. This approach is not straightforward if the open-reading frame is in the middle of an operon, which is the case for the ytgCR ORF. Allelic exchange in this configuration would require that additional sequences within and extending beyond the ytg operon would need to be included in the vector, which would likely exceed the current workable vector size allowed by Chlamydia transformation. Ultimately, a chromosomal mutation would not have addressed issues with low endogenous expression, even when tagged with FLAG.

We considered additional criteria with our use of the transformants in this experiment:

- *First is that the duration of anhydrous tetracycline (aTc) induction should be minimal, as we have observed that extended induction periods drastically reduce Chlamydial growth, i.e. smaller inclusions and presence of aberrant bodies (data not shown). It is for this reason that we chose an induction period of 3 hours, which did not have an obvious effect on the transformed strains under normal conditions. This observation prompted us to investigate the **inducibility (i.e. short-term induction)** of YtgCR from the plasmids, as opposed to effects on an artificial “steady-state” (i.e. long-term induction) level of expression induced by aTc. We believe that monitoring inducibility using short induction periods is better suited to the experimental system implemented here.*

- *Second, aTc induction should be sufficient to be detectable in Western blots and by immunofluorescence. We found that the protein could not be detected by Western blot despite the extended 6 hrs of induction with 200 nM aTc. In E. coli, clear bands were detectable when induced for only 3h with 100 nM aTc (Supplementary Fig. 4a-b). Prolonged induction periods were not suitable to answer our question because, as noted above, the transformants did not tolerate it well. We attempted to optimize this protocol in multiple ways, including immunoprecipitation of the protein via its FLAG tag and inhibition of the secreted chlamydial protease CPAF by either lysing cells in boiling SDS buffer (Snaveley et al, 2014) or by addition of the CPAF inhibitor lactacystin (Huang et al, 2008). Neither approach improved detection of YtgR-FLAG in our lysates, indicating that the relatively low level of expression from the pBOMBL vector (which contains a modified ribosome binding site to reduce leaky expression) combined with the poor tolerance to long aTc induction periods likely precluded detection by immunoblot. However, we were able to reliably detect specific immunofluorescent signal at sufficiently minimal induction times, and therefore we chose to optimize a condition in which we could quantify FLAG-specific fluorescent signal as a proxy for YtgCR expression.*
- *Third, we wanted to maintain the duration of tryptophan starvation comparable to that in Figure 2b for biological relevance. Because we could not detect YtgCR-FLAG by Western blot from total infected cell lysate (Supplementary Fig. 4b), we quantified fluorescence signals from YtgCR-FLAG and GroEL, the latter being tryptophan poor and constitutively expressed throughout chlamydial development, which made it an ideal control. We found that a 16-hour tryptophan starvation was sufficient, hence our decision to present that data. The transformant-based approach was sensitive enough to detect a difference in YtgCR^{YF}-FLAG and YtgCR^{YF}-FLAG without requiring 24 hours of tryptophan starvation as shown in Fig. 2b.*
- *An additional reason for quantifying immunofluorescence was the potential issue of heterogeneity in protein levels between inclusions, which we believed at the start would be a disadvantage of using Western blot. Heterogeneity in expression from a plasmid is typical, and the reason for this remains undefined. The plasmid we used is a derivative of the endogenous plasmid, which is maintained at low-copy numbers. Quantifying fluorescence at the level of individual inclusions gave us some idea of the level of heterogeneity, but nevertheless allowed us to detect differences between the two versions of YtgCR at shorter tryptophan starvation. Overall, we felt that our choice to quantify immunofluorescence was justified, and better than Western blot.*
- *Furthermore, in response to the concerns of the reviewers, we have created novel transformants of the YF and WWW mutants to express the respective proteins with a 3x-FLAG tag at the C-terminus instead of a single FLAG tag. We reasoned that this may increase sensitivity for western blot detection and allow us to rely on this technique to verify our findings. As you may already know, obtaining transformants is not a trivial approach in Chlamydia. Nevertheless, we were able to obtain verified clones to use in Western blots. These strains were used in our tryptophan starvation experiments designed to demonstrate that tryptophan dependency of YtgCR levels is mediated by the WWW motif. Efforts to optimize detection from these constructs included prolonged periods (6h+) of high aTc concentrations (200 nM), which was not well tolerated by the bacteria. We were unable to observe any difference in expression under control treatments (e.g. AN3365) or tryptophan-starved conditions, including indolmycin treatment (see figure below). We concluded that the induction conditions required to reliably detect YtgR-3xFLAG by immunoblot precluded the observation of any effects arising from starvation conditions.*

Left: Immunoblot of YtgR-3xFLAG in the presence or absence of 1.0 $\mu\text{g}/\text{mL}$ AN3365 (AN). Treatment began at 16 hpi and proceeded for 8h in the presence of 200 nM aTc. We expected reduction in expression of both WWW and YYF version in the presence of the AN.

Center: Immunoblot of YtgR-3xFLAG in the presence or absence of 240 μM indolmycin (IND). Treatment began at 16 hpi and proceeded for 8h in the presence of 200 nM aTc. Indole was supplemented to parallel indolmycin-treated cultures to rescue YtgR-3xFLAG through out-competing indolmycin by chlamydial tryptophan biosynthesis via TrpBA. We expected to observe specific reduction in the expression of WWW-YtgR-3xFLAG in the presence of indolmycin, and that this reduction could be rescued by the addition of indole. Contrastingly, we expected the YYF-YtgR-3xFLAG to be relatively unaffected.

Right: Immunoblot of YtgR-3xFLAG in the presence or absence of tryptophan. Treatment began at 8 hpi and at 16 hpi 200 nM aTc was added for an additional 8h. Indole was supplemented to parallel tryptophan-starved cultures to rescue YtgR-3xFLAG expression through replenishing the tryptophan pool by chlamydial tryptophan biosynthesis via TrpBA. We expected to observe specific reduction in the expression of WWW-YtgR-3xFLAG in the presence of indolmycin, and that this reduction could be rescued by the addition of indole. Contrastingly, we expected the YYF-YtgR-3xFLAG to be relatively unaffected.

- In an effort to alleviate concerns over the quantification of the immunofluorescent FLAG signal, we have done the following: first, we have replotted the data in Fig. 2f-e as a dot plot to hopefully better reflect the unique downward shift in FLAG signal intensity in the WWW population under Trp-starved conditions as compared to that of the YYF population. These plots include error bars representing the median absolute deviation, which is an error estimate comparable to the standard deviation from the mean. We feel that these changes help reflect the magnitude of effect, despite population heterogeneity. Second, we have provided histograms of the data that have been binned to show that a substantially larger portion of the Trp-starved WWW population bins into a lower fluorescence signal pool than the unstarved WWW or either YYF population. Again, we hope that this helps reinforce that the effect size is robust and specific to the WWW population.

The third approach, illustrated in Fig. 3 was the ectopic expression of the two YtgCR versions in *E. coli* to demonstrate the effect of indolmycin inhibition of tryptophanyl-tRNA synthetase, and underscore the role of translation at the triple *trp* codons in mediating the effects of tryptophan starvation. *E. coli* was a more suitable system to perform these experiments for the following reasons:

- Conditions for indolmycin treatment suitable to produce a physiological effect on chlamydia are optimized to induce persistence. While persistent chlamydia may show an effect on YtgR expression, we cannot solely attribute decreased levels to translation across the codons encoding the WWW motif. Indolmycin treatment of *E. coli* is straightforward, and as shown in Fig. 3c-d, indolmycin effects on YtgR expression was linked to the presence of the motif. *E. coli* are much more sensitive to these treatment conditions, requiring only a few hours of treatment to produce detectable effects on translation.
- To perform similar assays in *Chlamydia* would require the use of our transformants, which as noted above, do not appear to be amenable to these treatment conditions even under conditions where we can detect ectopic YtgR expression. The observation that 8h of AN3365

treatment does not produce a noticeable effect on ectopic expression of the constructs in Chlamydia despite almost completely ablating expression in E. coli with just two hours of treatment suggests that there are inherent barriers to implementing these tools similarly in Chlamydia.

We hope that this clarifies our rationale for using three different approaches. We felt that we had to adjust our approaches to fit the context, i.e. infection, use of transformants, and Chlamydia-related technical limitations of using indolmycin/AN3365. We have edited the text for this section of the manuscript to include a description of our validation of the transformants that necessitated the conditions utilized (Lines 298-311; 331-33; 339-41).

Issue 4: What is gained from the confocal microscopy in Fig. 2c? Localization isn't mentioned?

The microscopy in Figure 2c-d is meant to provide representative images from which the data in Figure 2e-f is derived.

Again, this does not answer the question of what is gained from doing confocal microscopy to quantify expression of a protein.

Please see our response to Issue 3 above regarding quantification of YtgCR levels by fluorescence intensity vs. Western blotting. For clarity, we would like to reemphasize that a particular advantage of using immunofluorescent microscopy to measure FLAG signal from individual inclusions in these assays is to account for inter-inclusion heterogeneity, which is expected to be a substantial factor with the use of plasmid-based chlamydial transformants experiencing stress that will have inter-cellular heterogeneity (i.e. media-defined tryptophan starvation). This point was made in response to Reviewer 2 Issue 3 in the first revision.

Issue 5: Line 318 the statements about YtgCR cleavage in E. coli in are confusing. It's not clear why this is relevant nor why it is different in minimal medium, and it isn't clear as compared to what other medium.

The sentences now read, "As in Ctr, YtgCR expressed in E. coli grown in rich media is cleaved and typically yields multiple cleavage products, suggesting non-specific cleavage due to dysregulated or off-target proteolytic activity. However, in minimal media, we observed that cleavage of YtgCR yielded one major band at the expected molecular weight of YtgR (30 kDa)."

Why would proteolysis be different in rich media vs minimal? I still find this an odd topic to be focused on in this paper and at best it is distracting and doesn't really add to the discussion of the mechanism of attenuation.

The main purpose of including this passage is to explain discrepancies in the blots that we published a few years ago (Thompson et al. 2012), which showed more cleavage bands than what is in Figure XX. This appeared to be correlated to bacterial growth in rich vs. minimal media. We do not know the reason for this at this time. The purpose of including this was to preempt any questions from readers of this manuscript and our published one (Thompson et al. 2012) that may arise. Having said this, we have now removed the corresponding text so as not to detract from the main points.

Reviewer #2 (Remarks to the Author):

Issue 1: The revised manuscript and rebuttal have addressed a number of points from the initial review. I have just a few unresolved issues about the different assays to measure YtgR levels, which were a concern for both reviewers. The authors provided explanations for the selective use of Western blots vs IFA assay, but there remains suspicion that the IFA assay was used because the Western blots were not sensitive enough to show a difference.

*We refer the Reviewer to our response to **Issue 3** from Reviewer 1.*

1. The IFA assay results remain unconvincing. There was a wide range of signal strength, with almost complete overlap between Trp+ and Trp- conditions for both YtgR and the GroEL control, and modest difference in the population median. What was the fold change in the population mean, and was this statistically significant? Why were violin plots used in Fig 2e-f but boxplots in Supp Fig 4c-d?

The fold change in the mean between Trp+ and Trp- was 1.34 for YtgCR and 1.21 for YtgCR^{YYF}. Parametric tests on these data are not suitable because the distributions are not normal nor are they comparably distributed, which is why we chose to statistically analyze them using the non-parametric Wilcoxon rank sum test. However, another method for the analysis of non-parametric data is the permutation test, which randomly samples the data over dozens of iterations and computes a statistic indicating any difference in the two populations. For YtgCR, a permutation test shows a statistically significant difference with a p-value of 0.02154 between Trp+ and Trp-. For YYF, the test shows a non-significant difference with a p-value of 0.06343. It is important to note that the small difference in population means does not capture the difference in population distribution (better represented by the population median), which is clearly shifted downward in the Trp- WWW population. Indeed, the median fold change is 1.95 for WWW, and 1.28 for YYF.

We have changed the aTc induction figure to dot-plots as represented in Figure 2e-f

2. The YtgR Western blot with tryptophan limitation should have better normalization. The text describes YtgR and OmpA downregulation compared to GroEL_1, as well as unhindered GroEL_1 expression. But the latter effect cannot be seen since YtgR levels were normalized to GroEL_1. Levels of these 3 proteins should be normalized to another protein.

GroEL_1 was chosen because it is constitutively expressed, i.e. not developmentally regulated, and it is tryptophan-poor, which would make it relatively unresponsive to our tryptophan depletion protocol. In Figure 2b, it is clear that GroEL_1 did not yield the same response pattern as YtgR or MOMP. Having said this, we have also included a normalization of GroEL_2 band intensity to GroEL_1 band intensity to demonstrate that GroEL_2 expression is not comparably reduced by tryptophan limitation. As with GroEL_1, GroEL_2 is tryptophan poor and constitutively expressed, making it a suitable negative control for this analysis. We additionally note that GroEL (a.k.a Hsp60/cHsp60) is commonly used in the field as a loading control for infected samples to indicate both the presence/status of infection and the comparable loading of chlamydial protein (see for example Siegl et al, 2014; Pennini et al 2010; Gonzalez et al 2014; Al-Zeer et al 2017; Beatty et al 1994).

REVIEWERS' COMMENTS

Reviewer #1 (Remarks to the Author):

The manuscript is much improved. The introduction is better at giving the reader a perspective about the importance of this organism and its significance for human health. It also makes the rest of the paper more understandable.

The use of several different methods to assay essentially the same phenomenon, while still a bit awkward, is at least partially rationalized.

As I have said on both prior reviews, this is an interesting system and adds a novel regulatory mechanism to those which control tryptophan biosynthesis in bacteria.

Hence I now recommend that this manuscript be published in Nature Communications

Reviewer #2 (Remarks to the Author):

The revised manuscript has been improved in 3 ways:

1. Clarified the description of transcriptional termination of the trp operon in *E. coli*
2. Provided explanations for why 3 different methods were used to measure YtgR protein levels
3. Improved the data presentation (dot plots for immunofluorescence studies, and normalization of the YtgR Western blot in response to tryptophan limitation)

In the rebuttal, the authors included additional Western blots of transformed strains expressing YtgR-3xFLAG. In these strains, YtgR was now detectable with anti FLAG antibodies, unlike with the YtgR-FLAG strains, although there was no difference in YtgR levels with tryptophan depletion. These data should be added to the Supplementary figures to show that the limitation of the Western blot method is that the difference in YtgR levels is small and not because the protein cannot be detected.

REVIEWERS' COMMENTS

Reviewer #1 (Remarks to the Author):

The manuscript is much improved. The introduction is better at giving the reader a perspective about the importance of this organism and its significance for human health. It also makes the rest of the paper more understandable.

The use of several different methods to assay essentially the same phenomenon, while still a bit awkward, is at least partially rationalized.

As I have said on both prior reviews, this is an interesting system and adds a novel regulatory mechanism to those which control tryptophan biosynthesis in bacteria.

Hence I now recommend that this manuscript be published in Nature Communications

We thank the reviewer for their constructive feedback during the review. We feel that it has substantially improved the final manuscript.

Reviewer #2 (Remarks to the Author):

The revised manuscript has been improved in 3 ways:

1. Clarified the description of transcriptional termination of the trp operon in E. coli
2. Provided explanations for why 3 different methods were used to measure YtgR protein levels
3. Improved the data presentation (dot plots for immunofluorescence studies, and normalization of the YtgR Western blot in response to tryptophan limitation)

In the rebuttal, the authors included additional Western blots of transformed strains expressing YtgR-3xFLAG. In these strains, YtgR was now detectable with anti FLAG antibodies, unlike with the YtgR-FLAG strains, although there was no difference in YtgR levels with tryptophan depletion. These data should be added to the Supplementary figures to show that the limitation of the Western blot method is that the difference in YtgR levels is small and not because the protein cannot be detected.

The immunoblots provided in the response have now been incorporated into Supplementary Figure 4 and referenced in the relevant section of the manuscript to explain that while we could detect YtgR-3xFLAG, the induction conditions prevented us from observing changes in protein expression, necessitating our use of immunofluorescent confocal microscopy.

We also thank Reviewer 2 for their constructive feedback throughout the review process.